# CoA synthase regulates mitotic fidelity via CBP-mediated acetylation

Chao-Chieh Lin [1], Mayumi Kitagawa[2], Xiaohu Tang[3], Ming-Hsin Hou[4], Jianli Wu[1], Dan Chen Qu[1], Vinayaka Srinivas[2], Xiaojing Liu[5], J. Will Thompson[5,6], Bernard Mathey-Prevot[5,7], Tso-Pang Yao[5], Sang Hyun Lee[2] & Jen-Tsan Chi[1]

The temporal activation of kinases and timely ubiquitin-mediated degradation is central to faithful mitosis. Here we present evidence that acetylation controlled by Coenzyme A synthase (COASY) and acetyltransferase CBP constitutes a novel mechanism that ensures faithful mitosis. We found that COASY knockdown triggers prolonged mitosis and multinucleation. Acetylome analysis reveals that COASY inactivation leads to hyper-acetylation of proteins associated with mitosis, including CBP and an Aurora A kinase activator, TPX2. During early mitosis, a transient CBP-mediated TPX2 acetylation is associated with TPX2 accumulation and Aurora A activation. The recruitment of COASY inhibits CBP-mediated TPX2 acetylation, promoting TPX2 degradation for mitotic exit. Consistently, we detected a stage-specific COASY–CBP–TPX2 association during mitosis. Remarkably, pharmacological and genetic inactivation of CBP effectively rescued the mitotic defects caused by COASY knockdown. Together, our findings uncover a novel mitotic regulation wherein COASY and CBP coordinate an acetylation network to enforce productive mitosis.

[1] Department of Molecular Genetics and Microbiology, Center for Genomic and Computational Biology, Duke University School of Medicine, Durham, NC 27710, USA. [2] Program in Cancer and Stem Cell Biology, Duke-NUS Medical School, 8 College Road, Singapore 169857, Singapore. [3] Department of Biological Sciences, Michigan Technological University, Houghton, MI 49931, USA. [4] Department of Medicine, National Yang-Ming University, Taipei 11217, Taiwan. [5] Department of Pharmacology and Cancer Biology, Duke University School of Medicine, Durham, NC 27710, USA. [6] Duke Proteomics and Metabolomics Shared Resource, Center for Genomic and Computational Biology, Duke Cancer Institute, Duke Molecular Physiology Institute, Duke University, Durham, NC 27710, USA. [7] Department of Pediatrics, Duke University, Durham, NC 27708, USA. Correspondence and requests for materials should be addressed to J.-T.C. (email: jentsan.chi@duke.edu)

T he eukaryotic cell cycle is a highly regulated and orchestrated process. The precise temporal and spatial progression of mitosis is tightly regulated by protein phosphorylation and ubiquitination, which control the levels and activities of proteins crucial for mitotic progression[1]. The microtubules are rearranged to form a bipolar spindle that functions to segregate a duplicate set of genetic materials into two daughter cells[2]. During early mitosis, Targeting Protein for Xklp2 (TPX2) is one critical regulator of many aspects of the spindle assembly, including nucleating microtubules around chromosomes, targeting mitotic proteins to mitotic spindles and activating Aurora A kinase[3–6]. Activated Aurora A then triggers a phosphorylation cascade that determines the precise timing of mitotic spindle assembly and disassembly[7,8]. During metaphase to anaphase transition, the anaphase-promoting complex/cyclosome mediates the ubiquitination and degradation of TPX2, thereby terminating Aurora A kinase activity for proper mitotic exit[3]. Anomalous activation of Aurora A can lead to aberrant mitosis and contribute to genomic instability commonly overserved in cancer[9–11]. However, the regulatory mechanisms that govern the precise window of TPX2 expression, demarcated by its sharp increase and rapid decline during mitosis, remain mostly uncharacterized.

Reversible histone acetylation has been intensively studied as a key epigenetic mark. The abundance and distribution of histone acetylation plays a critical role in chromatin structure central to gene expression[12]. CREB-binding protein (CBP) is one of the best characterized histone acetyltransferases. By catalyzing histone acetylation, CBP acts as a transcriptional co-activator and facilitates gene transcription. CBP is also known to have non-genomic functions and non-histone substrates[13]. CBP has been shown to stabilize multiple non-histone proteins by acetylation[14,15].

Additionally, reversible acetylation is intimately linked to metabolism[16]. This connection is thought, at least in part, to be bridged by acetyl-CoA, the common source of acetyl group for both lysine acetylation and metabolic flux. The biosynthesis of Coenzyme A (CoA) requires CoA synthase (COASY), a bifunctional metabolic enzyme catalyzing the last two steps of de novo Coenzyme A biosynthesis[17]. In human, the R499C mutation of COASY has been reported to disrupt COASY enzymatic activity, leading to neurodegeneration[18]. In *Schizosaccharomyces pombe* and *Drosophila melanogaster*, CoA biosynthesis has been suggested to maintain DNA integrity and proper mitosis[19,20]. However, whether COASY plays any role in the regulation of protein acetylation or mitosis remains unclear.

Here we report the identification of COASY as a novel regulator of mitosis via its stage-specific interaction with CBP during mitosis to inhibit CBP acetyltransferase activity. We found that CBP acetylates and stabilizes TPX2 to promote Aurora A activation during mitosis. Importantly, at later stage of mitosis, the physical binding of COASY to CBP interferes with CBP-mediated TPX2 acetylation, thereby promoting Aurora A inactivation and proper mitotic exit. Therefore, COASY knockdown led to persistent TPX2 protein levels and prolonged Aurora A activation, resulting in mitotic defects. These findings identify an important contribution of TPX2 acetylation, regulated through the interaction of COASY and CBP, to proper regulation of mitotic progression and prevention of genome instability.

## Results

### COASY knockdown induced mitotic defects.
In a genetic screen, we serendipitously identified that COASY knockdown reduced the viability of a triple-negative breast cancer cell line (MDA-MB-231) (Supplementary Fig. 1a, b). COASY is a metabolic enzyme required for the last two steps of de novo CoA biosynthesis[17]. Consistent with the role of COASY in CoA synthesis, we verified that COASY knockdown reduced the level of CoA and acetyl-CoA using LC-MS analysis (Supplementary Fig. 1c, d). However, the acetyl-CoA/CoA ratio was not altered (Supplementary Fig. 1e). Surprisingly, we also found that COASY knockdown altered cell morphology into a flattened cobbledstone appearance (Fig. 1a) and a multinucleation phenotype (Fig. 1b). We found that COASY knockdown similarly triggered multinucleation in multiple cancer cell lines, including A549 (lung adenocarcinoma) (Fig. 1c), MDA-MB-231 (breast cancer) (Supplementary Fig. 1f), PANC-1 (pancreatic cancer) (Supplementary Fig. 1g), and non-cancerous cell line ARPE-19 (human retina) (Supplementary Fig. 1h). Since multinucleation often results from aberrant mitotic progression[21], we used a live cell imaging approach to determine how COASY knockdown affected the mitotic progression (Fig. 1d, Supplementary Movies 1 and 2). We found that COASY knockdown extended mitosis from an average of 38 to 162 min (Fig. 1d, e) and significantly increased the occurrence of cytokinesis failure from 2.3% to ~60% (Fig. 1f). To further confirm the extended mitosis, we synchronized A549 cells at the prometaphase using thymidine-nocodazole block. The expression of cyclin B1, a mitotic marker, was then examined after releasing the cells into fresh media (Fig. 1g). In control cells, cyclin B1 protein mostly disappeared 40 min after thymidine-nocodazole release, indicating the normal duration of mitosis. In contrast, in COASY knockdown cells, cyclin B1 protein remained detectable up to ~160 min after release (Fig. 1g). Together, these results indicate that COASY knockdown prolonged mitosis and caused cytokinesis failure.

### COASY knockdown increases acetylation of specific proteins.
Given that CoA is the main acetyl carrier for the reversible acetylation of lysine residues in proteins, we speculated that COASY knockdown would affect the acetylation of proteins, leading to mitotic defects. Therefore, we performed an acetylome analysis in synchronized A549 cells treated with control or COASY siRNAs. We identified a total of 1074 acetylated peptides that belong to 504 proteins (1% false discovery rate (FDR)) (Supplementary Fig. 2a, b and Supplementary Table 1). Two-tail Student's *t*-test identified 119 differentially acetylated peptides (from 96 proteins) between the control and COASY-knockdown samples with a cutoff value of $p < 0.001$ (Fig. 2a and Supplementary Table 2). Although the lower acetyl-CoA level associated with COASY knockdown would logically predict reduced protein acetylation, to our surprise, we found that COASY knockdown increased the acetylation in 105 peptides (9.8%) of total acetylated peptides (1074 peptides) while only reduced the acetylation in 14 peptides (1.3%) of total acetylated peptides (Supplementary Fig. 2b and Supplementary Table 2). By analyzing the hyper- and hypo-acetylated proteins using STRING database[22], we found that the hyperacetylated proteins form a strong protein association network with significant amount of experimental determined interactions (Fig. 2b). Interestingly, we found that CBP acetyltransferase is a prominent node with a large number of connections with other hyperacetylated proteins, suggesting its potential importance to these hyperacetylated proteins. We also identified clusters of proteins involved in mitosis/cytokinesis, including TPX2, TOP2A, SMC3, CENPF, KIF23, ANLN, and TXN. Of note, COASY knockdown led to the hyperacetylation of SMC3 K106, a modification previously implicated in regulating sister chromatid cohesion[23,24]. Moreover, many members of the HBO1 acetyltransferase complex, including KAT7, PHF15, PHF16, MEAF6, ING4, and BRPF3, were found as a cluster[25]. In contrast, the hypoacetylated proteins showed only weak or no interaction (Supplementary Fig. 2c).

**TPX2 is a downstream effector of COASY during mitosis**. Given the prominent mitotic phenotypes associated with COASY knockdown, we focused on the candidate proteins involved in mitosis. Among the hyperacetylated peptides, we found a notable 5–10 fold enrichment in three acetylated peptides (containing either K75, K476, or K582) of the TPX2 protein (Fig. 3a and Supplementary Fig. 3a–c), a key co-activator of Aurora A kinase[4,5]. By comparing the TPX2 sequences among 15 different species, we found that K476 and K582, but not K75, were evolutionarily conserved (Supplementary Fig. 3d). To validate the increased acetylation on TPX2 protein under COASY knockdown during mitosis, we enriched the mitotic population of A549 cells by thymidine-nocodazole block and mitotic shake-off. The levels of TPX2 protein (Fig. 3b), but not mRNA (Supplementary Fig. 3e, f), were markedly higher in cells with COASY knockdown. TPX2 was then pulled down by TPX2 antibody and probed with a pan-

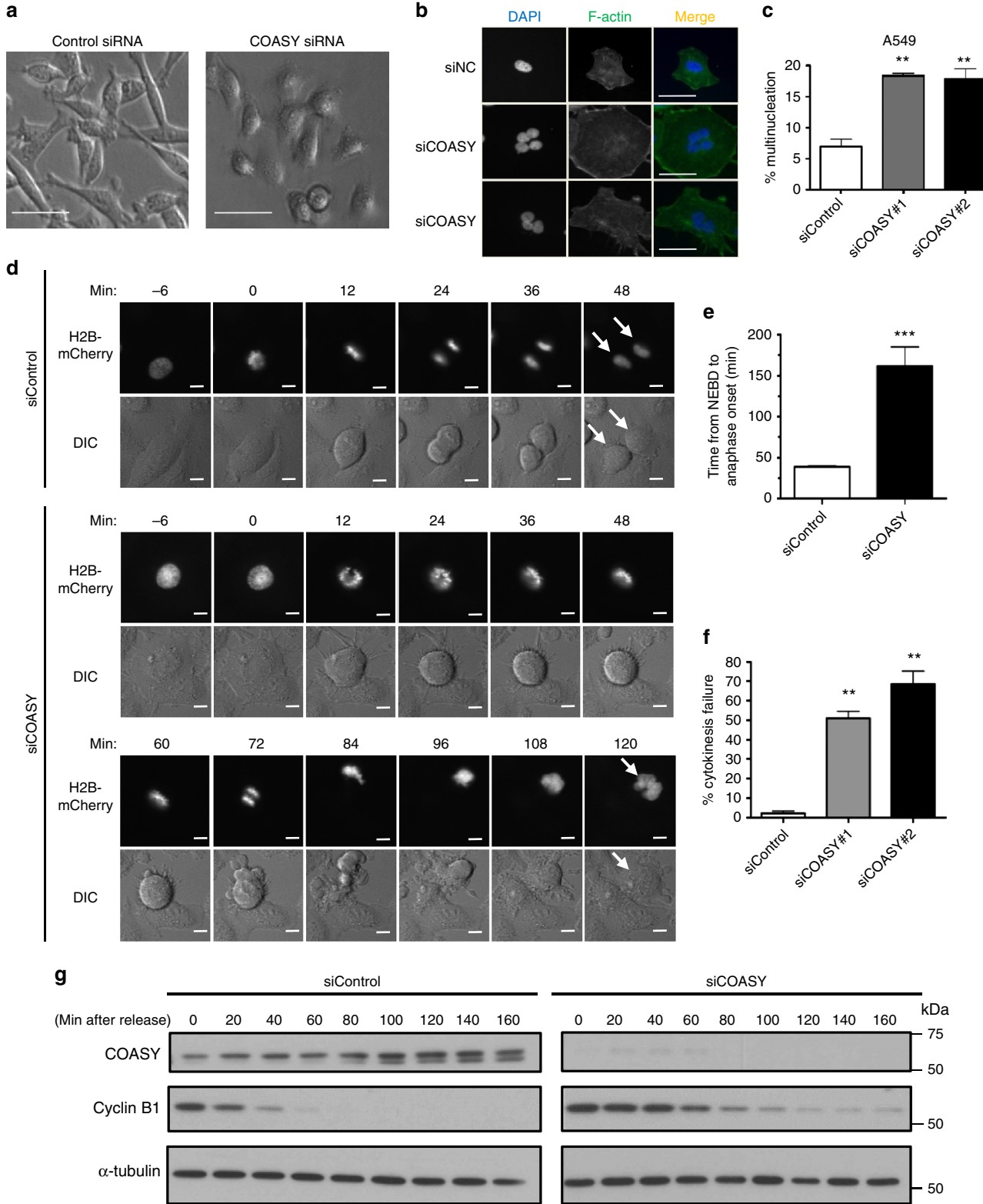

acetylated lysine antibody. When normalized to the TPX2 protein levels, COASY knockdown indeed increased TPX2 acetylation (Fig. 3c). The increase in acetylation at K476 and K582 upon COASY knockdown was further validated and quantified in a site-specific manner by parallel-reaction monitoring (PRM)[26] targeted mass spectrometry using synthetic heavy isotope peptides (Supplementary Fig. 3g–l). Since protein lysine acetylation could regulate protein stability by affecting protein ubiquitination via competing for similar sets of lysine residues[27], we investigated whether COASY knockdown affected TPX2 ubiquitination. We transfected V5-tagged TPX2 cDNA into HEK-293T cells treated with either control or COASY siRNA. TPX2 was then immunoprecipitated by V5 antibody for western blots. With MG132 treatment, a proteasome inhibitor, we found that while COASY

knockdown did not affect the general ubiquitination levels (Fig. 3d, left panel), it markedly reduced TPX2 ubiquitination (Fig. 3d, right panel). These findings suggest that the increased TPX2 acetylation induced by COASY knockdown may prevent TPX2 ubiquitination and increase its levels.

Since the induction and degradation of TPX2 is tightly regulated during the cell cycle to control mitotic spindle and Aurora A kinase activity[28], the aberrant mitosis phenotype in COASY knockdown might be contributed by dysregulated TPX2 protein level. Therefore, we knocked down the expression of TPX2 and COASY simultaneously. We found that the multi-nucleation (Fig. 3e) and extended mitosis phenotype (Supplementary Fig. 3m, n) induced by COASY knockdown was significantly suppressed by simultaneous knockdown of TPX2.

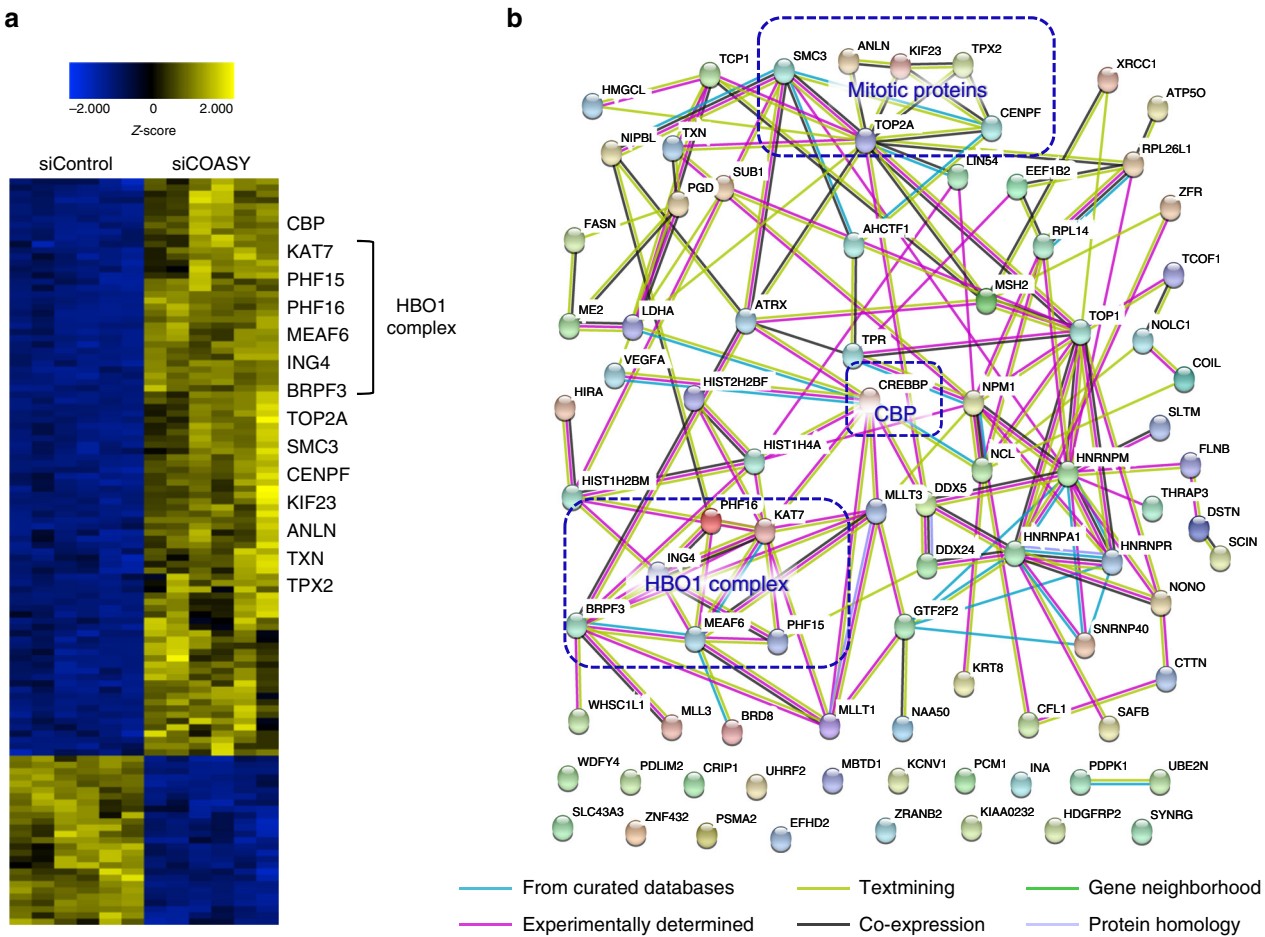

**Fig. 2** COASY knockdown leads to hyper-acetylation of a protein network. **a** Heatmap and hierarchical clustering of a subset of selected peptides whose levels have been significantly altered in A549 cells by COASY knockdown ($n = 119$); $p < 0.001$ (two-tailed Student's $t$-test). **b** Functional protein association network of hyperacetylated proteins under COASY knockdown generated using STRING database

**Fig. 1** COASY knockdown induced multinucleation and prolonged mitosis. **a, b** COASY knockdown by siRNA in MDA-MB231 triggered cobble-stone morphological change (**a**) and multinucleation (**b**). MDA-MB-231 cells transfected with control or COASY siRNA for 72 h were stained with DAPI (nuclei) and Alexa Fluor 488 phalloidin (F-actin). Scale bars, 50 mm (**a**), 20 mm (**b**). **c** The percentage of multi-nucleated A549 cells after transfection with control or two independent COASY siRNAs for 72 h. For each sample, more than 150 cells were examined by immunofluorescence microscopy. **d** Live cell time-lapse imaging showed that COASY knockdown extended mitosis and induced cytokinesis failure. A549 cells expressing histone 2B (H2B)-mCherry (nucleus marker) were transfected with control or COASY siRNA for 24 h before live cell imaging. Scale bars, 10 mm. **e** COASY knockdown increased the time in mitosis as determined by the time of nuclear envelope breakdown (NEBD) to anaphase onset (in min). For each sample, more than 35 cells were examined by live cell imaging. **f** COASY knockdown increased percentage of cytokinesis failure during mitosis. For each sample, more than 35 cells were examined by live cell imaging. **g** Western blots show that COASY knockdown extend the time of mitosis by using cyclin B1 expression as a mitotic marker. A549 cells were synchronized by thymidine-nocodazole block and released in fresh media. The samples were then harvested every 20 min and analyzed by western blots. Bars show standard error of the mean. **p < 0.01, ***p < 0.001, two-tailed Student's $t$-test, $n = 3$ independent repeats

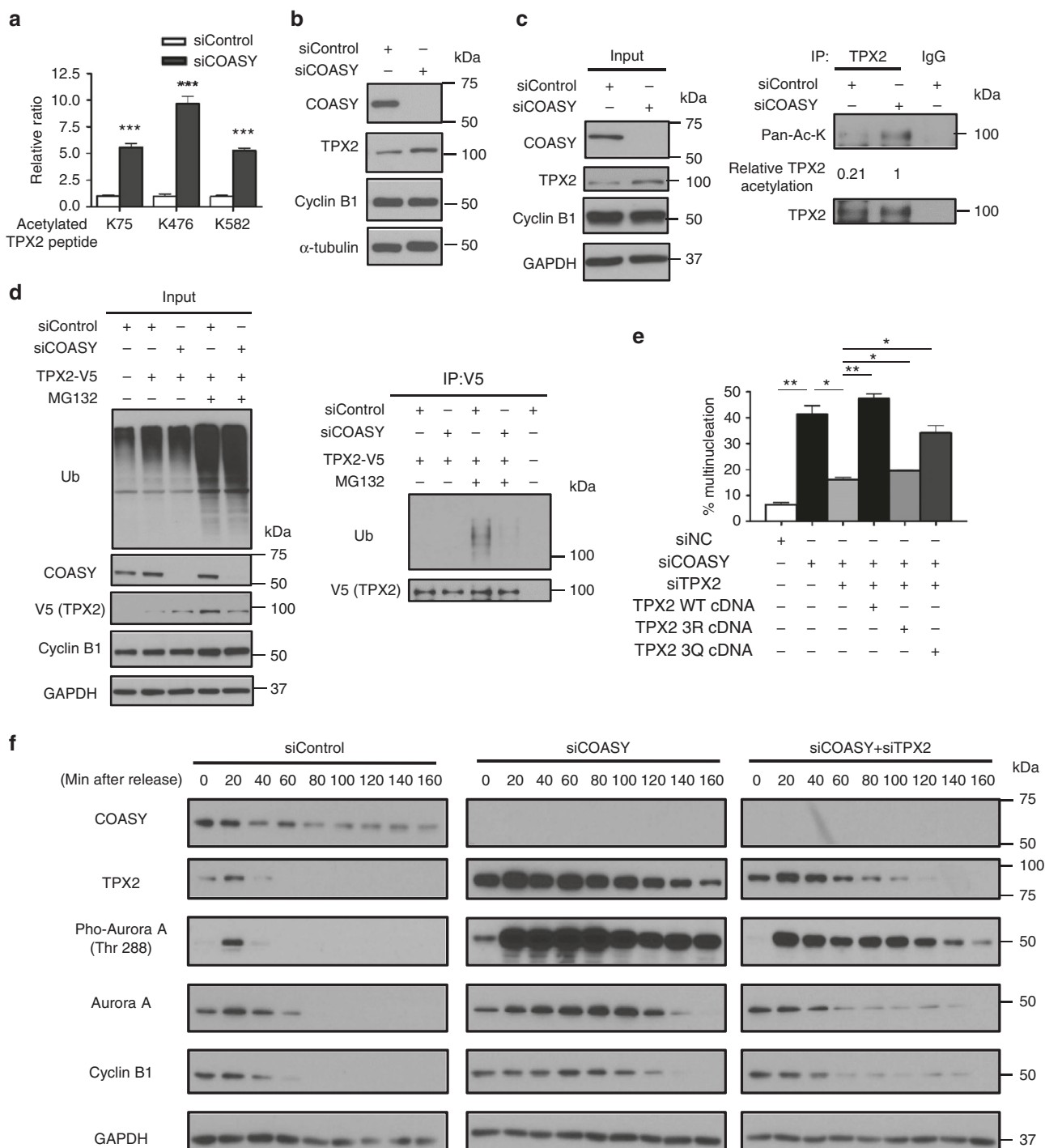

**Fig. 3** TPX2 is the downstream effector of COASY leading to multinucleation and extended mitosis. **a** COASY knockdown increased three acetylated peptides (K75, K476, and K582) of TPX2 as determined by acetylome analysis. **b** COASY knockdown increased TPX2 protein level in A549 cells. **c** COASY knockdown increased the TPX2 acetylation in A549 cells. A549 cells enriched in early mitosis by thymidine-nocodazole block were transfected with control or COASY siRNA and probed with indicated antibodies The TPX2 in the cell lysates were then immunoprecipitated and blotted for pan-acetylated lysine antibody. **d** COASY knockdown decreased TPX2 ubiquitination in HEK-293T cells. HEK-293T cells were transfected with COASY siRNA and TPX2 cDNA. After MG132 treatment, TPX2 were immunoprecipitated and blotted for pan-ubiquitination antibody. **e** TPX2 protein expression level and acetylation on TPX2 regulated the percentage of COASY-dependent multinucleation. TPX2 knockdown rescued the multinucleation induced by COASY knockdown. Reintroducing wild type (WT) or acetylation-mimetic mutant (3Q), but not acetylation-deficient (3R), of siRNA-resistant TPX2 cDNA recapitulated multinucleation phenotype. Multinucleation was determined as described in the legend to Fig. 1. **f** COASY knockdown triggered increased Aurora A Thr 288 phosphorylation and extended mitosis by the elevation of TPX2. A549 cells were synchronized by thymidine-nocodazole block and released in fresh media for the indicated time (in min). COASY knockdown increased TPX2 expression and delayed its decline during from 40 min (control) to >160 min. COASY knockdown also delayed the inactivation of Aurora A and degradation of cyclin B1, indicating extended mitosis. The extended mitosis caused by COASY knockdown can be abolished by simultaneous TPX2 knockdown. Bars show standard error of the mean. $*p < 0.05$, $**p < 0.01$, $***p < 0.001$, two-tailed Student's $t$-test, $n = 3$ independent repeats

Under the same experimental design, we further re-expressed siRNA-resistant wild-type TPX2, acetylation-deficient (3R) or acetylation-mimicking (3Q) TPX2 mutants cDNA at close to physiological level (Supplementary Fig. 3n). We found that wild-type and acetylation-mimicking mutant TPX2 expression increased multinucleation and extended mitosis, while acetylation-deficient mutant significantly reduced these phenotypes (Fig. 3e and Supplementary Fig. 3m, n). These data indicated that the elevation and lysine acetylation of TPX2 protein contributes to the mitotic defects induced by COASY knockdown.

Next, we determined the role of TPX2 mis-regulation in the mitotic defects induced by COASY knockdown. A549 cells transfected with control, COASY siRNA or COASY siRNA combined with TPX2 siRNA were synchronized by thymidine-nocodazole block and released. Samples were collected at different time points and used to evaluate the levels and activities of TPX2 and Aurora A kinase by western blots (Fig. 3f). In the control cells, TPX2 protein level increased sharply and reached a peak after 20 min that is accompanied by the autophosphorylation (Thr288) of Aurora A kinase. After 40 min, TPX2 protein level, Aurora A phosphorylation and Cyclin B1 all plummeted, indicating mitotic exit (Fig. 3f, left panel). Remarkably, COASY knockdown enhanced TPX2 protein level and Aurora A phosphorylation in both amplitude and duration; the mitotic exit was significantly prolonged to ~120 min as indicated by the persistence of cyclin B1 (Fig. 3f, middle panel). When TPX2 was suppressed to approximately physiological level using siRNA (Fig. 3f, right panel), the prolonged mitosis induced by COASY knockdown was significantly rescued as indicated by the levels of cyclin B1 and Aurora A phosphorylation (Fig. 3f, right panel).

Since TPX2 level and Aurora A phosphorylation were upregulated under COASY knockdown (Fig. 3f), we further examined the localization of these signals by confocal microscopy (Supplementary Fig. 3o–q). Under COASY knockdown, we observed stronger and broader distribution of TPX2 proteins reaching from spindle poles to chromosomes during metaphase when comparing to the control (Supplementary Fig. 3o). During metaphase, the stage Aurora A are both activated under both control and COASY knockdown, COASY knockdown did not dramatically change the signal of Aurora A phosphorylation (Supplementary Fig. 3p). However, during interphase, Aurora A phosphorylation was much higher under COASY knockdown, indicating COASY can determine the precise activation of Aurora A at specific stages of cell cycle (Supplementary Fig. 3q). Taken together, these results strongly suggested that TPX2 dysregulated by COASY knockdown contributed to the abnormal TPX2 distribution, Aurora A activation and prolonged mitosis.

**CBP acetylates and stabilizes TPX2 protein**. Next, we investigated how COASY knockdown increases TPX2 acetylation. In the COASY-dependent acetylome, we found that COASY knockdown increased the acetylation of two over-lapping peptides that include K1762 of the acetyltransferase CBP (Fig. 4a and Supplementary Fig. 4a, b). As autoacetylation on CBP is known to increase its acetyltransferase activity[29] and CBP showed extensive interactions with other hyperacetylated proteins in our acetylome dataset (Fig. 2b), we hypothesized that CBP might be responsible for increased TPX2 acetylation and protein accumulation under COASY knockdown. To test this possibility, we first examined whether CBP forms a complex with TPX2. By using TPX2 antibody to pull down the endogenous TPX2 protein, we observed that endogenous CBP was co-immunoprecipitated independent of COASY (Supplementary Fig. 4c). Next, we tested whether CBP acetylated TPX2. In HEK-293T cells, V5-tagged TPX2 was co-transfected with either wild type or a catalytic deficient mutant

HA-CBP construct[30] (Fig. 4b). When co-transfected with wild type, but not catalytic mutant CBP, wild type CBP was co-immunoprecipitated with acetylated TPX2. (Fig. 4b). Similar results were also repeated in MDA-MB-231 cells (Supplementary Fig. 4d). Since TPX2 acetylation appears to negatively correlated with its ubiquitination (Fig. 3c, d), we tested whether CBP-mediated TPX2 acetylation contribute to the stabilization of TPX2. We co-transfected a V5-TPX2 construct with either wild type or catalytic deficient mutant CBP into HEK-293T cells and determined the half-life of V5-TPX2 protein in the presence of cycloheximide, an inhibitor of protein synthesis (Fig. 4c). When co-transfected with mutant CBP, the half-life of TPX2 protein was significantly decreased to ~3 h comparing with wild-type CBP (Fig. 4c). Similar trends were also observed in A549 cells (Supplementary Fig. 4e). These results indicated that the acetylation modification of TPX2 is CBP-dependent and contributes to TPX2 protein accumulation.

To further test whether TPX2 is acetylated by CBP on the three lysine residues (K75, K476, and K582) enriched in our COASY knockdown acetylome dataset (Figs. 2a and 3a), the V5-tagged TPX2 or TPX2-RRR (acetylation-deficient mutant) were co-transfected with HA-tagged CBP in HEK-293T cells. The V5-TPX2 protein was then pulled down by V5 antibody and detected using a pan-acetylated antibody. TPX2-RRR showed an ~90% reduction in acetylation, indicating the acetylation modification of K75, K476, and K582 are the main lysine residues on TPX2 acetylated by CBP (Supplementary Fig. 4f). To further confirm whether the three lysine residues affect TPX2 ubiquitination, we overexpressed V5-tagged TPX2 or TPX2-RRR under MG132 treatment. The TPX2 protein in cell lysates was pulled down by V5 antibody and detected using a pan-ubiquitination antibody, which detect a broad range of ubiquitylated TPX2 species from 100 to 160 kDa (Supplementary Fig. 4g, h). We found a decrease in TPX2 ubiquitination in TPX2-RRR under both native (HEK-293T cells, Supplementary Fig. 4g) and denaturing conditions (MDA-MB-231 cells, Supplementary Fig. 4h), indicating these three acetylation sites are crucial for TPX2 ubiquitination.

To directly assess the role of CBP in the TPX2 upregulation and multinucleation upon COASY knockdown, we inhibited CBP by either siRNA or a chemical inhibitor C646 (Fig. 4d and Supplementary Fig. 4i)[31]. Remarkably, inhibition of CBP by either method significantly rescued the aberrant upregulation of TPX2 and multinucleation phenotype in COASY knockdown cells (Fig. 4d, e and Supplementary Fig. 4i, j). Therefore, CBP contributed to both TPX2 upregulation and the multinucleation in COASY knockdown scenario.

We then test whether CBP is a physiological regulator of the TPX2 protein levels during mitosis under normal condition in synchronized A549 cells (Fig. 4f and Supplementary Fig. 4k). By suppressing CBP expression (by siRNA) or activity (by C646), we found that CBP inhibition markedly blunted the sharp TPX2 increase during early mitosis (Fig. 4f and Supplementary Fig. 4k). This finding indicates that CBP regulates TPX2 protein levels during normal mitosis.

Since CBP and p300 have similar sequences and function[32], we further tested whether p300 is also involved in COASY-dependent multinucleation. We knocked down the p300 by targeting its specific sequences together with COASY (validated by RT-PCR, Supplementary Fig. 4l, m). We did not observe a rescue of the multi-nucleation phenotypes (Supplementary Fig. 4n). This finding indicates that CBP, but not p300, is involved in COASY-dependent mitotic defects.

**COASY inhibits CBP-mediated TPX2 acetylation**. To further characterize the interacting network among COASY, CBP, and TPX2 in vivo, we examined their subcellular localizations during

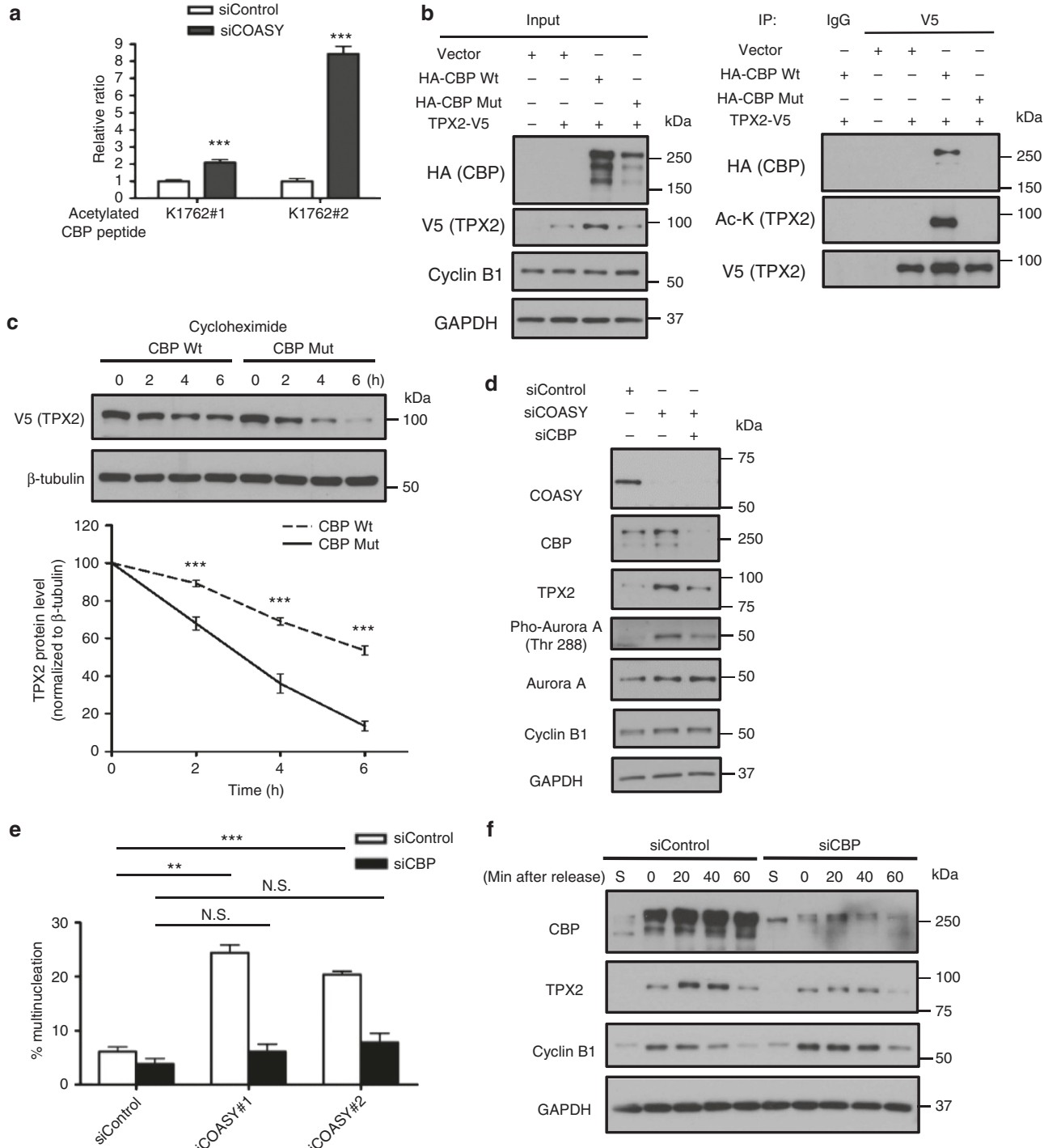

**Fig. 4** CBP acetylates and stabilizes TPX2 protein level. **a** COASY knockdown increased two CBP-acetylated peptides that cover K1762. **b** CBP-acetylated TPX2 in vivo. Wild type or catalytic-deficient mutant of HA-tagged CBP was cotransfected with V5-tagged TPX2 into HEK-293T cells. After 20 min of release from thymidine-nocodazole block, the V5-tagged TPX2 was then immunoprecipitated with control (IgG) or V5 antibody (V5) and then probed with HA (CBP), acetylated lysine or V5 (TPX2). Wild type, but not catalytic-deficient mutant, CBP increased TPX2 acetylation and levels. **c** CBP increased the half-life of TPX2. Wild type or catalytic-deficient mutant of CBP was cotransfected with V5-tagged TPX2 into unsynchronized HEK-293T cells. After 24 h, the protein synthesis of the transfected cells was halted with 25 mg/ml cycloheximide and collected at the indicated times for western blots. The TPX2 protein level at indicated time points was quantified by Image J software and normalized to b-tubulin protein level. **d** CBP siRNA abolished the increased TPX2 and Aurora-A phosphorylation caused by COASY knockdown. A549 cells transfected with control, siCOASY or siCBP were released from thymidine-nocodazole block for 20 min and probed for western blots using indicated antibodies. The phosphorylated Thr 288 of Aurora A indicates activation. **e** CBP knockdown rescued the multinucleation caused by COASY knockdown. Multinucleation was determined by immunofluorescence. **f** CBP knockdown decreased TPX2 protein level during mitosis. A549 cells transfected with siRNA targeting CBP were synchronized by thymidine-nocodazole block and released in fresh media. The released cells were harvested every 20 min and probed with indicated antibodies. To examine TPX2 protein level in S phase, additional samples were harvested after 24 h of thymidine block. Bars show standard error of the mean. **a**, **e** N.S. not significant ***$p < 0.001$, **$p < 0.01$, two-tailed Student's $t$-test, $n = 3$ independent repeats. **c** Two-way ANOVA: $p < 0.0001$. Bonferroni post hoc tests, ***$p < 0.001$, $n = 4$ independent repeats

mitosis. TPX2 is known to be localized to spindle microtubules during mitosis[33], but the locations of CBP and COASY during mitosis has not been reported. We expressed a GFP-tagged COASY in A549 cells and examined the subcellular localizations of COASY, TPX2, and CBP in different stages of cell cycle by confocal microscopy (Fig. 5a, b and Supplementary Fig. 5a). The stage of cell cycle for each cell was defined by the shape of nuclei (DAPI staining). During interphase, COASY showed a diffuse pattern mainly in nucleus and cytoplasm similar to the pattern of COASY from the Human Protein Atlas[34]. However, the subcellular localization of COASY showed no obvious colocalization with TPX2 or CBP (Fig. 5a, b). Remarkably, during mitosis, COASY-GFP, but not GFP, is recruited to spindle microtubules and co-localized with TPX2 and CBP (Fig. 5a, b and Supplementary Fig. 5a). These findings support the idea that TPX2, CBP, and COASY interact with each other during mitosis. To further examine the physical interactions between COASY, TPX2, and CBP during mitosis, A549 cells synchronized in prometaphase by thymidine-nocodazole block were released into fresh media and harvested at different time points. The endogenous CBP and associated proteins were immunoprecipitated by a CBP antibody followed by western blots (Fig. 5c). Both COASY and TPX2 showed a stage-specific interaction with CBP, and these interactions largely disappeared by 80 min (Fig. 5c). Reciprocally, the endogenous COASY in mitotically synchronized HEK-293T cells also associated TPX2 and CBP proteins indicated by co-immunoprecipitation (Supplementary Fig. 5b). Since the physical interaction of CBP and COASY is novel and unexpected, we further validated this interaction by pulling down CBP from HEK293T cells overexpressing Flag-CBP and COASY cDNA. By using mass spectrometry, we confirmed the enrichment of COASY peptides when CBP was immunoprecipitated (Supplementary Fig. 5c–f). These data strongly suggest that COASY, CBP, and TPX2 physically associate with each other in a specific stage of mitosis.

In principle, COASY could inhibit CBP by direct binding to CBP or by the help of some other unidentified proteins in COASY–CBP-TPX2 protein complex in vivo. Therefore, we tested whether COASY could directly inhibit CBP acetyltransferase activity on TPX2 in vitro. First, when recombinant CBP (catalytic domain) and TPX2 were incubated with acetyl-CoA, we noted a significant increase in the acetylation of TPX2 (Fig. 5d). When wild-type TPX2 was replaced with acetylation-deficient mutant protein, acetylation signal became undetectable (Supplementary Fig. 5g). Next, we employed the TPX2 acetylation assay to determine the effects of COASY. Importantly, purified recombinant COASY inhibited CBP-mediated TPX2 acetylation in a dose-dependent manner (Fig. 5e, f and Supplementary Fig. 5d). Together, these data suggest that COASY can directly inhibit CBP-mediated TPX2 acetylation, consistent with the observed CBP hyperacetylation upon COASY knockdown.

**The PPAT domain regulates TPX2 protein stability**. COASY contains two catalytic domains: the phosphoribosyl pyrophosphate amidotransferase (PPAT) domain and the Dephospho-CoA kinase (DPCK) domain (Fig. 6a)[17]. These two domains are responsible for the two sequential enzymatic steps required for the CoA synthesis. In humans, the disease-causing R499C mutation of COASY disrupts the enzymatic activity of DPCK[18]. We therefore tested whether this R499C mutation affects the association of COASY with CBP and leads to mitotic abnormalities. We found that the interaction between COASY and CBP was not affected by R499C mutation on COASY (Supplementary Fig. 6a). Interestingly, we found that COASY with R499C mutation was able to rescue the multinucleation phenotype

caused by COASY knockdown comparable to wild type COASY (Fig. 6b and Supplementary Fig. 6b). These results suggested that the enzymatic activity of DPCK of COASY was not essential for mitotic regulation. To further distinguish which domain of COASY contributes to the multinucleation phenotype, we generated V5-tag expression constructs for the three functional domains in COASY protein (N terminus, PPAT and DPCK) and compared their ability to rescue the multinucleation under COASY knockdown (Fig. 6c). While all three constructs result in comparable protein expression based on V5 tag (Fig. 6d, left panel), only the PPAT, but not N-terminal or DPCK, was able to rescue the multinucleation defects induced by COASY knockdown (Fig. 6c). Next, we compared the ability of these domains in their physical interaction with CBP by co-immunoprecipitation assay. We immunoprecipitated HA-CBP and probed with different COASY domains using V5 antibody. Consistent with multinucleation rescue phenotype (Fig. 6c), we found that only PPAT, but not N-terminal and DPCK, strongly interacted with CBP (Fig. 6d, right panel). We next examined whether the reintroduction of intact COASY or PPAT, DPCK domain could disrupt CBP-mediated TPX2 protein stabilization previously shown in Fig. 4c (Fig. 6e–g and Supplementary Fig. 6c). Full-length COASY, PPAT, or DPCK domain was cotransfected with V5 tagged TPX2 and CBP cDNA in HEK-293T cells. After 24 h of incubation, TPX2 expression was translationally inhibited by cycloheximide. The efficiency of CBP-mediated TPX2 protein stabilization was then assessed by monitoring TPX2 degradation. Overexpression of full-length COASY successfully reverse the CBP-dependent TPX2 stabilization (Fig. 6e). Intriguingly, PPAT, but not DPCK domain, was sufficient to disrupt TPX2 stabilization (Fig. 6f, g). Taken together, these data indicate that PPAT domain of COASY associates with CBP and regulates TPX2 protein stabilization to ensure mitotic fidelity.

COASY encodes the last enzyme in the multi-step steps during the de novo synthesis of CoA from panthenate acid (Vitamin B5)[17]. To further determine the role of CoA synthesis pathway in the multi-nucleation phenotypes observed with COASY knockdown, we used siRNAs to inhibit other upstream enzymes in the CoA biosynthesis pathway, including pantothenate kinases (PANKs) and phosphopantothenoylcysteine decarboxylase (PPCDC) (Supplementary Fig. 6d–g). Neither PANKs nor PPCDC siRNAs triggered TPX2 upregulation or similar multi-nucleation phenotypes seen with COASY knockdown (Supplementary Fig. 6h, i). Collectively, the data suggests that COASY regulates CBP activities and mitotic fidelity through a mechanism independent of CoA levels. However, we still cannot completely rule out the relevance of enzymatic activities of COASY in its regulation of CBP.

**Low COASY expression is associated with poor outcome**. Next, we wish to determine the functional and pathological relevance of COASY expression. We have provided evidence that COASY knockdown triggered CBP-mediated TPX2 upregulation and persistent activation of Aurora A. Given that overexpression of Aurora A was reported to confer resistance to chemotherapeutic agent taxol[9], we speculated that low COASY expression can lead to taxol resistance. To test this possibility, we transfected control or COASY siRNA in MDA-MB-231 cells and exposed these cells to a dose-titration of taxol. Indeed, we found that COASY siRNA led to significant resistance of MDA-MB-231 cells to taxol (Supplementary Fig. 7a). Next, we examined whether COASY expression in primary tumors was associated with different clinical outcome in breast cancer cohorts using prognostic database PROGgene V2[35]. We found that lower level of COASY in primary tumor was associated with poor clinical outcomes in the

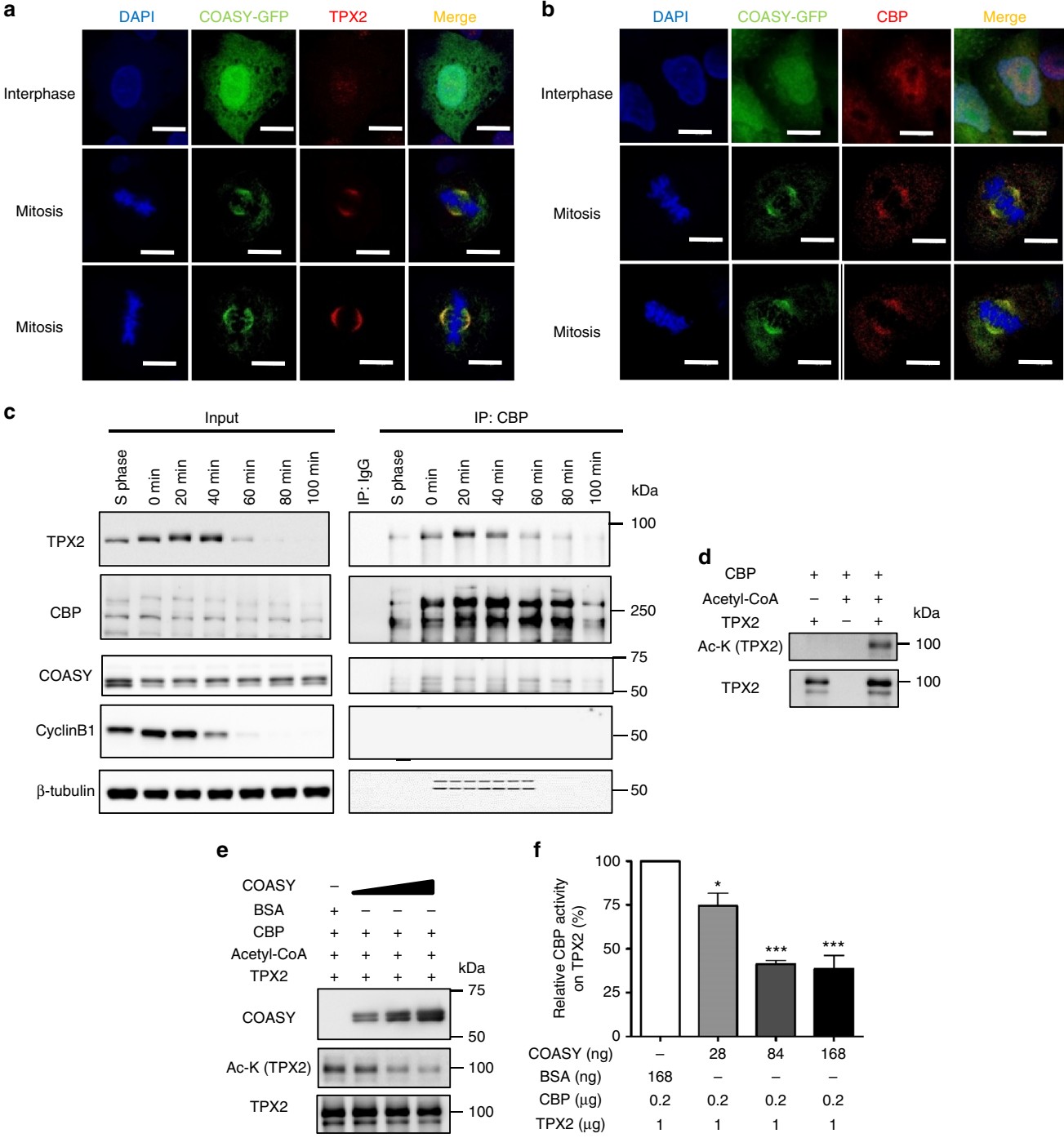

**Fig. 5** Stage-specific physical association and in vitro CBP activity assays suggest direct interaction among COASY, CBP, and TPX2. **a** TPX2 and **b** CBP showed colocalization with COASY in spindle microtubules during mitosis. COASY-GFP expressing A549 cells were stained with TPX2 (**a**) or CBP (**b**) antibody for confocal microscopy. Scale bars, 10 mm. **c** TPX2 and COASY physically associate with CBP during mitosis. A549 cells were synchronized by thymidine-nocodazole block and released for the indicated time to enrich cells in different cell cycle stages. An additional sample was harvested after 24 h of thymidine block for cells enriched in S phase. The CBP was immunoprecipitated, separated by electrophoresis and probed with the TPX2 and COASY antibodies. **d** CBP-acetylated TPX2 in vitro. Recombinant full-length TPX2 protein was acetylated by CBP (catalytic domain) human recombinant protein in the presence of acetyl-CoA. Acetylation on TPX2 was determined by western blots using pan-acetylated lysine antibody. **e** COASY inhibited CBP-mediated TPX2 acetylation in vitro. Bacterially purified COASY or control BSA was added to the CBP activity assay using TPX2 protein as substrate. With increasing amount of COASY, the level of acetylated TPX2 decreased in a dose-dependent manner. **f** Quantification of TPX2 acetylation in the presence of COASY. Acetylated TPX2 level determined by pan-acetylated lysine antibody was quantified by Image J software and normalized to TPX2 protein level. One-way ANOVA: $p < 0.0001$. Bonferroni post hoc tests, *$p < 0.05$, ***$p < 0.001$; $n = 4$ independent repeats. Bars show standard error of the mean

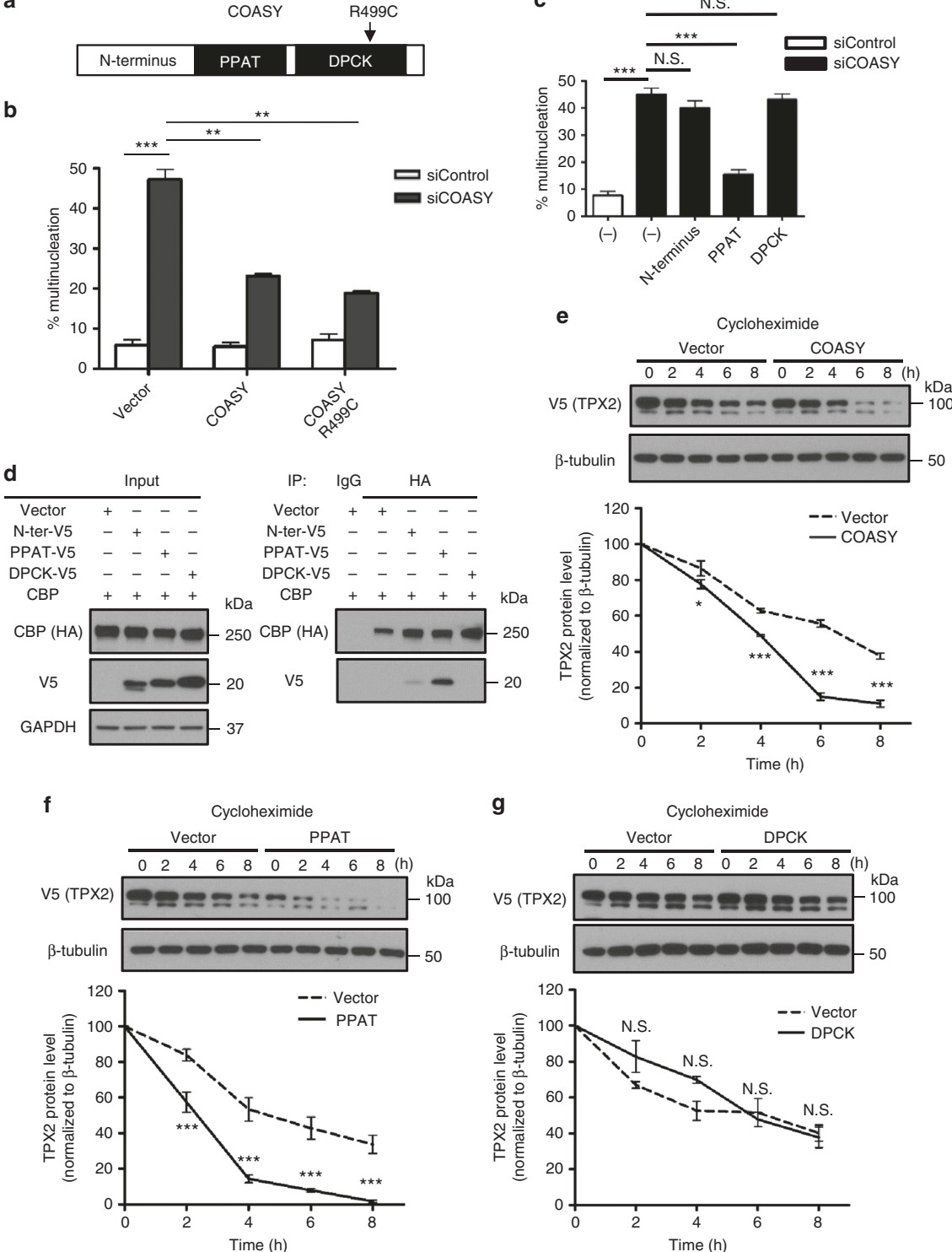

**Fig. 6** PPAT on COASY regulates TPX2 protein stability by interacting with CBP. **a** Schematic illustration of domains on COASY. COASY is composed of N terminus regulatory domain and two catalytic domain (PPAT and DPCK). R499C mutant inactivates DPCK. **b** Both wild and R499C of COASY cDNAs rescued multinucleation. MDA-MB-231 cells stably expressing control vector (pLKO.1), wild type or R499C COASY were transfected with COASY siRNA for 72 h. The multinucleation induced by COASY siRNA can be abolished by both wild type and R499C COASY. **c** Overexpression of PPAT rescued the multinucleation induced by COASY knockdown. *$p < 0.05$, **$p < 0.01$, two-tailed Student's $t$-test, $n = 3$ independent repeats. **d** PPAT and CBP showed strong interaction. The cDNA of N-terminus domain, PPAT or DPCK were cotransfected with CBP cDNA to HEK-293T cells. The cells were then enriched in mitosis by nocodazole treatment and harvested for co-immunoprecipitation. **e–g** Both COASY and PPAT, but not DPCK, promote the degradation of TPX2 protein. COASY (**e**), PPAT domain (**f**), DPCK domain (**g**) was cotransfected with CBP and V5-tagged TPX2 into unsynchronized HEK-293T cells. After 24 h, the protein synthesis of the transfected cells was halted with 25 mg/ml cycloheximide and collected at the indicated times for western blots. The TPX2 protein level at indicated time points was quantified by Image J software and normalized to b-tubulin protein level. **b, c** N.S. not significant ***$p < 0.001$, **$p < 0.01$, two-tailed Student's $t$-test, $n = 3$ independent repeats. **e–g** Two-way ANOVA: $p < 0.0001$ (**e**), $p < 0.0001$ (**f**), N.S. (**g**). Bonferroni post hoc tests, *$p < 0.05$, ***$p < 0.001$, $n = 3$ independent repeats. Bars show standard error of the mean

Enerly and Steinfeld dataset (GSE19783)[36] (Supplementary Fig. 7b). The Loi dataset (GSE6532)[37] and Jonsdottir dataset (GSE46563)[38] also revealed that lower expression of COASY was correlated with decreased time to metastases (Supplementary Fig. 7c, d). Together, these data suggest the strong correlation of reduced COASY expression level with taxol resistance and poor clinical outcome of cancer patients.

Collectively, our results strongly support that the stage-specific COASY–CBP association during mitosis regulates CBP-dependent TPX2 protein acetylation and abundance and leads to precise modulation of Aurora A kinase and other mitotic components, eventually resulting in proper mitotic progression (Supplementary Fig. 8). In early mitosis, CBP facilitates the TPX2 acetylation to reduce ubiquitination and stabilizes TPX2. At mitotic exit, COASY is relocalized to spindle microtubules and associates with CBP to inhibit acetyl-transferase activities. The stage-specific CBP inactivation destabilizes TPX2 and results in the sharp decline of TPX2 level to allow mitotic exit. Thus, COASY knockdown leads to the CBP overactivation of CBP, hyper-acetylation and accumulation of TPX2, resulting in the mitotic defects and multi-nucleation.

Together, the experimental data and proposed model provides a conceptual framework of stage-specific COASY-CBP disassociation and association regulates the precise timing of TPX2 accumulation and degradation during mitosis.

## Discussion

In mitosis, protein acetylation has been an under-characterized layer of regulation compared to protein phosphorylation and ubiquitination. Acetylation on the lysine residues during mitosis have been reported on histones[39], cohesion[40], and the spindle checkpoint protein BubR1[41]. Interestingly, our acetylome analysis upon COASY knockdown revealed a distinct set of hyper-acetylated mitotic regulatory proteins, including histone acetyltransferase CBP. Indeed, our data supported a crucial role of overactivation of CBP in the dysregulated TPX2 levels and mitotic defect phenotypes caused by COASY knockdown. While CBP is shown to cooperates with APC/C to promote transcription and protein degradation during mitosis[42], much remains unknown about the regulation and role of the catalytic activities of CBP during mitosis. Our data revealed an unexpected role of CBP, together with COASY, in regulating mitotic progression, at least in part, through orchestrating a tight temporal accumulation and degradation of TPX2 to ensure mitotic fidelity.

During mitosis, the expression of TPX2 is tightly regulated where it promptly accumulates during early mitosis and then undergoes rapid degradation towards mitotic exit[3]. Our data have revealed a novel regulatory mechanism that the interaction between COASY and CBP cooperatively regulate the initiation and exit of mitotic cycle. During early mitosis, we found that CBP-mediated TPX2 acetylation is important for the TPX2 accumulation. Acetyl-CoA is the essential substrates for CBP-mediated acetylation and the levels of acetate and acetyl-CoA are proposed to serve as energy gauge to allow proliferation[43,44]. The CBP-mediated TPX2 acetylation and accumulation may be one crucial mechanism by which acetyl-CoA levels are coupled with cellular proliferation. Our results indicate that the sub-cellular localization of COASY is cell cycle-regulated and recruited to spindle microtubules and co-localizes with CBP to regulate its activities toward TPX2. These data indicate that COASY is spatially and temporally regulated to control CBP activities and its downstream target proteins during mitosis. We surmise that, during interphase, COASY is involved in de novo CoA synthesis to support the adequate levels of acetyl-CoA in the cells. However, during mitosis, COASY contributes to the TPX2

degradation essential for mitotic exits by moving to spindle microtubules to associate with CBP and TPX2 to inhibit CBP-mediated TPX2 acetylation. Together, these data suggest a novel mean by which the TPX2 acetylation and levels are regulated by CBP activities and COASY localization during different stages of cell cycle.

CoA biosynthetic pathway consists of multiple catalytic steps that convert pantothenate to CoA. First, the pathothenate is phosphorylated by pathothenate kinase, followed by the addition of cysteine, followed by decarboxylation to form 4′-phospho-pantetheine. COASY catalyzes the last two steps of converting 4′-phosphopantetheine to CoA through its PPAT and DPCK domain to generate CoA. Mis-sense mutations in PANK2 and COASY can lead to neurodegeneration with brain iron accumulation in human[18]. Further study suggests that inhibition of pathothenate kinase can decrease the availability of CoA and lower the acetylation of specific subset of proteins[45]. Intuitively, knockdown of COASY should reduce the production of Coenzyme A, lower acetyl-CoA level, and cause global protein hypoacetylation. However, several recent studies suggest that most histone acetyltransferases have similar affinity for CoA and acetyl-CoA. Therefore, CoA can competitively inhibit acetyl-CoA production. Therefore, the acetyl-CoA/CoA ratio, instead of absolute acetyl-CoA levels, may be more relevant determinants of protein acetylation[46,47]. In our analysis of COASY-knocked down cells, while both CoA and acetyl-CoA are lower (Supplementary Fig.1c, d), the acetyl-CoA/CoA ratio remain unchanged (Supplementary Fig. 1e). Consistently, our acetylome analysis revealed that the majority (89%, 955 acetylated/1074 total) of the acetylated peptides were not significantly altered by COASY knockdown (Supplementary Fig. 2b). Therefore, COASY knockdown did not lead to global hypo-acetylation. Instead, COASY knockdown altered the acetylation of a specific set of proteins, including CBP and TPX2. These results are consistent with another report that decreasing CoA level in Drosophila or mammalian cells by inhibition of PANK, the upstream enzyme of COASY in CoA biosynthesis pathway, also did not alter the global pattern of protein acetylation[45]. Thus, our results are not inconsistent with other studies on the relationship between the level of protein acetylation with cellular CoA and acetyl-CoA abundance.

Instead, our acetylome analysis revealed that COASY knockdown, while reducing cellular CoA and acetyl-CoA, predominantly leads to hyperacetylation of a small subset (9.8%) of total acetylated peptides, including hyperacetylated CBP associated with increased enzymatic activities. We found that COASY controls acetylation, at least in part, by binding and inhibiting acetyltransferase CBP instead of affecting substrate levels. First, the knockdown of COASY, but not other enzymes in the CoA synthesis pathways, leads to increased TPX2 proteins and multi-nucleation (Supplementary Fig. 6h, i). Second, the enzymatic-deficient mutant of DPCK domain of COASY did not affect the ability to abolish the multi-nucleation phenotypes (Fig. 6b). Therefore, COASY is likely to regulate CBP through direct interactions instead of affecting substrate levels/availability for CBP. However, we cannot fully exclude the possible involvement of COASY catalytic activity in regulating CBP, even though the production of CoA per se is not responsible for CBP regulation and multi-nucleation phenotypes. In the future, the structural and functional analysis of COASY will be critical to elucidate the underlying mechanisms of CBP regulation during mitosis.

While the current study focuses on TPX2, other hyper-acetylated proteins induced by COASY knockdown are likely to be important for additional regulations of mitosis and cytokinesis. For example, the acetylation cycles of the cohesin complex protein SMC3 at K106 is required for proper sister chromatid cohesion and separation[23,24]. Therefore, the hyperacetylation of

SMC3 K106 under COASY knockdown may hinder the de-attachment of kinetochore during mitosis. Similarly, anillin is localized to the mid-body and critical for the cytokinesis. The hyperacetylation of anillin may reduce the efficiency of cytokinesis, further enhancing the mitosis defects associated with COASY knockdown. Although future study would be required to assess the impact of these hyperacetylation events, our studies have uncovered a mitotic acetylation network controlled by COASY and CBP. During mitosis, nuclear envelope breakdown and chromosome condensation render transcription inactive[48–51]. Our findings point out the importance of post-translational modification by COASY-modulated CBP in regulating TPX2 protein and potentially other mitotic proteins found in our acetylome analysis. Since many mitotic and cytokinesis proteins become hyper-acetylated upon COASY knockdown, a broad, coordinated and reversible protein acetylation event may be required during mitosis. Our data strongly suggest a guardian role of COASY for a proper execution of mitosis.

## Methods

**Cell culture and plasmids**. MDA-MB-231 cells, A549 cells, PANC-1 cells, and APRE-19 cells were obtained from Duke Cell Culture Facility (Durham, NC, USA). As suggested by Duke Cell Culture Facility, all four cell lines were cultured in Dulbecco's modified Eagle's medium (DMEM; GIBCO-11995) supplemented with 10% fetal bovine serum and 1× antibiotics (penicillin, 10,000 UI/ml and strepto-mycin, 10,000 UI/ml). These cell lines have been validated to be mycoplasma-free and authenticated by STR DNA profiling before being frozen by the Duke Cell Culture Facility (Durham, NC, USA), and were maintained for fewer than 6 months. The cells were maintained in a humidified incubator at 37 °C and 5% $CO_2$. For COASY-GFP expressing cells, COASY cDNA from pDONR223-COASY was cloned into pLenti CMV GFP DEST (736–1) using Gateway cloning following the manufacturer's protocol. PLX302-TPX2 was generated by transferring TPX2 cDNA from pENTR223-TPX2 to PLX302 using gateway cloning. Lentivirus was generated by transfecting HEK-293T cells with a 1:0.1:1 ratio of pMDG2:pVSVG: pLKO.1 with TransIT-LT1 transfection reagent (Mirus). COASY-GFP expressing cell line were generated by adding 250 μl virus to a 60-mm dish of A549 cells with polybrene (8 μg/ml). pLenti CMV GFP DEST (736–1) was a gift from Eric Cam-peau (Addgene plasmid # 19732)[52]. PLX302 V5-N terminus domain (1–179), PLX302 V5-PPAT (180–350), PLX302 V5-DPCK (351–564) were generated by Gateway cloning. pDONR223-COASY was a gift from William Hahn & David Root (Addgene plasmid # 23660)[53]. pcDNA3β-FLAG-CBP-HA and pcDNA3β-FLAG-CBP-LD-HA were gifts from Tso-Pang Yao (Addgene plasmid # 32908)[54]. pLX302 was a gift from David Root (Addgene plasmid # 25896)[55]. pENTR223-TPX2 was a gift from The ORFeome Collaboration (DNASU). mEmerald-AuroraAKD-C-7 was a gift from Michael Davidson (Addgene plasmid # 54010).

**SiRNA and drug treatments**. SiRNA knockdown was performed using lipo-fectamine RNAiMAX (ThermoFisher) following the manufacturer's protocol. AllStars negative control siRNA (#1027281) was purchased from ThermoFisher. COASY siRNAs (D-006751-01, D-006751-02, D-006751-03), CBP siRNA (D-003477-18) and TPX2 siRNA (D-010571-04) were purchased from Dharmacon. To enrich the mitotic cells under COASY silencing, A549 cells were transfected by siRNA using lipofectamine RNAiMAX (Invitrogen) following the manufacturer's protocol. After 24 h of incubation, the cells were treated with 2 mM of thymidine (T1895; Sigma) for 16 to 18 h. After 2 h of release in fresh media, the cells were treated with 100 ng/ml nocodazole (M1404; Sigma) for 16 h. The cells were then released and harvested by mitotic shake-off for further experiments. C646 (5 mM, SML0002; Sigma) was added to the media at the same time point nocodazole was treated. For ubiquitination assay, cells overexpressing TPX2-V5 were treated with MG132 (M7449, Sigma) for 6 h. TPX2-V5 in the lysed samples were pulled down by V5 antibody (MA5-15253, ThermoFisher) and blotted with ub antibody (sc-8017, Santa Cruz).

**Measurement of CoA and Acetyl-CoA**. MDA-MB-231 cells were cultured in DMEM medium in a six-well plate. After 3 days of COASY siRNA knockdown, the medium was replaced by 80% methanol/water and incubated in −80 °C for 15 min to inactivate enzyme activity. After centrifugation at 20,000 rcf for 10 min at 4 °C. The supernatant was dried by speed vacuum at room temperature[56]. The meta-bolites were reconstituted into 30 μl of 50 mM ammonium acetate, and 10 μl was injected to LC-MS for acyl-CoA profiling analysis[5].

**Live-cell imaging**. H2B-mCherry was a gift from Robert Benezra (Addgene plasmid # 20972)[57]. A549 cells stably expressing H2B-mCherry were selected using G418 (800 μg/ml) for 14 days. The cells were then silenced by COASY or control siRNAs. The phenotypes of extended mitosis and cytokinesis failure were assessed

in living cells using Olympus VivaView FL incubator microscope. The images of DIC and mCherry were taken every 6 min.

**Immunofluorescence microscopy**. For AF 488 Phalloidin (A12379, Thermo-Fisher), CBP and TPX2 staining, cells were washed once with PBS and fixed in 3.7% paraformaldehyde for 15 min, followed by permeabilization and blocking with 0.2% Triton X-100 and 2% BSA for 15 min. Primary antibodies were incu-bated with the cells for 1 h. For images in Fig. 1a, b, transmitted light microscopy and immunofluorescence microscopy were performed using EVOS FL cell imaging system (ThermoFisher). For Fig. 4a, b, the confocal microscope (SP8, Leica) was used to detect and capture the signal of CBP, TPX2, and COASY-GFP.

**Acetylome analysis**. Global proteomics analysis of differential acetylation was performed in a manner similar to previously described, with relatively minor modifications[58]. Cell lysates were lysed in urea, digested with trypsin and enriched with pan anti-acetyl Lysine antibody. Enriched samples were analyzed by label-free liquid chromatography tandem mass spectrometry to determine the relative abundance of acetylated peptides between control and COASY knockdown con-dition. The raw data and spectrum files can be downloaded at ftp://massive.ucsd. edu/MSV000081937. For full description of the methods, see the Supplementary Information.

**Parallel-reaction monitoring**. Quantification of the native acetylated TPX2 pep-tides as a ratio to the stable-isotope internal standards was performed using PRM targeted mass spectrometry[26]. Using the combination of targeted mass spectro-metry and synthetic stable isotope labeled peptides, site-specific quantification of acetylation can be performed with little to no ambiguity. Skyline software (https:// skyline.ms) was utilized to export inclusion lists to the mass spectrometer and for quantitative data analysis. Samples were analyzed under the same LC conditions as described for discovery experiments, with the exception that a 60 min LC-MS gradient was used. The mass spectrometer utilized was a QExactive HF (Thermo Scientific), with precursor isolation window of 2.0 $m/z$, maximum accumulation of 0.5 s, target AGC of 1e6, and MS resolution of 60,000. The raw data and spectrum files can be downloaded at https://goo.gl/hy2fQd.

**Western blots and immunoprecipitation**. The cells were harvested and washed once with ice-cold PBS, then resuspended in NP-40 buffer with Trichostatin A, protease, and phosphatase inhibitors. The samples were lysed by incubating in 4 °C with constant vortex for 30 min, then spun down at 13,000 rpm for 10 min at 4 °C. Supernatant was transferred to new tube, and protein concentration was mea-sured by the Pierce BCA protein assay kit (#23225, ThermoFisher). Between 15 and 40 μg of protein was loaded on 8% SDS-PAGE gels, wet-transferred to PDVF membrane, blocked with 5% non-fat milk in 1× TBST, then incubated with pri-mary antibodies overnight at 4 °C. For immunoprecipitation, between 400 and 600 μg of protein lysate was incubated with 2 μg of primary antibody overnight at 4 °C. Dynabeads protein G was then added and incubated for 2 h. Beads were pulled down and boiled in 1× protein loading buffer for western blots. Antibodies: v5 (1:1000, MA5-15253, ThermoFisher), Cyclin B1 (1:2000, 4135s, Cell Signaling), GAPDH (1:2000, sc-25778, Santa Cruz), alpha-tubulin (1:2000, T9026, Sigma-Aldrich), HA (1:2000, sc-805, Santa Cruz), TPX2 (1;1000, 12245, Cell Signaling), TPX2 (1:1000, sc-376812, Santa Cruz), Aurora A (1:1000, 610938, BD Biosciences), Phospho-Aurora A pThr288 (1:1000, MA5-14904, ThermoFisher), COASY (1:1000, sc-393812, Santa Cruz), CBP (1:1000, sc-369, Santa Cruz), CBP (1:1000, 7389s, Cell Signaling). Acetylated lysine (1:1000, ICP0380, Immunechem). Acety-lated lysine (1:1000, 9441 s, Cell Signaling). Uncropped images of western blots are shown in Supplementary Fig. 9.

**Quantitative real-time PCR**. Total RNA was extracted by the RNeasy Mini Kit (Qiagen) following the manufacturer's protocol. RNA was reverse transcribed to cDNA by the SuperScript II (Invitrogen) using random hexamers. Quantitative real-time PCR was performed using Power SYBR Green PCR Mix (Applied Bio-systems) and StepOnePlus Real-time PCR system (Applied Biosystems) following the manufacturer's protocol. Beta-actin (reference gene) primers: sense, 5′-CAC TCTT CCA GCC TTC CTT C-3′, antisense, 5′-GGA TGT CCA CGT CAC ACT TC-3′; COASY primers: sense, 5′-TGT GGC TGA GGG AAA GCG T-3′, anti-sense, 5′-ACC TGG CGT TGG GTG ATA TG-3′; TPX2 primers: sense, 5′-ACT TCC GCA CAG ATG AGC G-3′, antisense, 5′-GGA TGC TTT CGT AGT TCA GAT GT-3′; PANK1 primers: sense, 5′-CTG CCT TGA TAA CCC ATA CCC T-3′, antisense, 5′-CTT GGA GTA CAC GGC TAG AAT G-3′; PANK2 primers: sense, 5′-AGG GGA CTA TGA GAG GTT TGG-3′, antisense, 5′-GCA CAC ATT CTT GCT ATT GAG AC-3′; PPCDC primers: sense, 5′-TGC CTC TTC TGG TGT CAA AGC-3′, antisense, 5′-TGT TTG GCT CTC TCA GTT GTG A-3′. Reactions were performed in triplicate.

**In vitro CBP activity assay**. Recombinant COASY or equal amount of control BSA were first combined with CBP protein in HAT buffer (50 mM Tris-base, pH 8.0, 10% glycerol, 0.1 mM EDTA, and 1 mM dithiothreitol) and incubated at 30 °C for 30 min. Acetyl-CoA and recombinant TPX2 protein were then added to the reactions and incubated at 30 °C for 30 min. Proteins were resolved by 10%

SDS–PAGE, and acetylation was detected by western blot using Acetyl lysine (1:1000, ICP0380, Immunechem). Recombinant proteins: CBP (03-189, Millipore), TPX2 (T40-30H-50, SignalChem).

**Statistical analysis**. Data represent the mean ± the standard error of the mean. $p$-values were determined by a two-tailed Student's $t$-test or ANOVA test with Bonferroni post hoc tests in Excel or Graphpad. Error bars represent SEM, and significance between samples is denoted as $*p < 0.05$; $**p < 0.01$; and $***p < 0.001$.

**Data availability**. All data supporting the findings of this study are available from the authors upon reasonable request.

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

## Acknowledgements

We are grateful for critical discussions and technical support from members of the Chi lab and Dr. Donald Fox; Dr. Suzanne Jackowski for providing Hopan; Ryan Annis and Dr. Mohanish Deshmukh for microinjection of CoA; Sarah Rains and Laura Dubois for technical assistance with mass spectrometry experiments. We acknowledge the financial support from the Pilot Project programs of Duke Cancer Institute, Duke/Duke-NUS Pilot Award, NIH (CA125618 to J.-T.C.), the Department of Defense (W81XWH-12-1-0148, W81XWH-14-1-0309, W81XWH-15-1-0486 to J.-T.C.). C.-C.L. is a Hung-Taiwan Duke fellow.

## Author contributions

C.-C.L. and J.-T.C. conceived the experiments and wrote the manuscript. C.-C.L. performed the majority of the experiments. J.-T.C., T.-P.Y. and S.H.L. supervised the work. M.K., X.T., M.-H.H., J.W., D.C.Q. and V.S. collaborated in the discussion and experiments. X.L. measured CoA and acetyl-CoA. J.W.T. performed acetylome and PRM. B.M.-P. provided siRNA library. B.M.-P., T.-P.Y. and S.H.L. provided critical feedback.

## Additional information

**Competing interests:** The authors declare no competing interests.

