## [Peer Review File · Nature Communications]

Reviewer #1 (Remarks to the Author):

The manuscript by Lin and colleagues presents data to suggest that Coenzyme A synthase (COASY) inactivation, perhaps paradoxically, leads to the hyper-acetylation of a network of proteins involved in mitotic regulation; COASY knockdown cells display prolonged mitoses and multinucleation. The authors do not offer any mechanistic insight into the relationship between COASY knockdown and hyper-acetylation, but focus their efforts on discerning the role of acetylation in mitotic progression. The authors present evidence to indicate that CBP acetylates and stabilizes the TPX2 protein, which in turn leads to the hyper-activation of Aurora A kinase, whose over-expression has previously been shown to promote cytokinesis failure and multinucleation (Cancer Cell, 2003 Jan;3(1):51-62.). In this regard the authors determine that COASY, CBP and TPX2 associate temporally at the mitotic spindle during mitosis and that the PPAT domain of COASY, which interacts with CBP, will rescue the multinucleation phenotype. The data presented is novel, and will be of interest to the wider scientific community.

Specific comments:

1. Although the authors present some evidence to suggest that TPX2 knockdown rescues the COASY knockdown-induced multinucleation phenotype and prolonged mitosis phenotype (Fig 3E and F), a specific role for the CBP-dependent acetylation of TPX2 in multinucleation and prolonged mitosis requires further exploration. The manuscript could be strengthened by testing the ability of an siRNA-resistant wild-type TPX2 and a TPX2 RRR mutant, which is resistant to CBP-directed acetylation, to rescue the prolonged mitoses, and/or multinucleation phenotype when the endogenous TPX2 species is knocked down by siRNA. It would also be interesting to see if a TPX2 QQQ acetylation mimic mutant, reinforces these phenotypes.
2. The authors present evidence of a TPX2 acetylation/ubiquitylation switch, such that the TPX2-RRR mutant which is unable to be acetylated by CBP, is also targeted for ubiquitylation (Figure S4). It appears that the ubiquitylation studies were performed by IP-WB, under native conditions. Although the results are suggestive of ubiquitylation of TPX2 at the K residues identified, these experiments should be repeated with His6-Ubiquitin under denaturing conditions to validate that it is TPX2 itself that is the ubiquitylation target.
3. Fig S1: Did siCOASY #3 affect acetyl-CoA levels or multinucleation?
4. Fig 3C. The increase in TPX2 acetylation by WB using a pan-AcK antibody following COASY knockdown is not convincing. It would be valuable to generate a site-specific AcK antibody for TPX2, against one, or more of the sites.
5. A number of experiments in this manuscript are performed with HEK-293T cells. It is well established that adenovirus E1A associates with CBP and p300 to reprogramme cellular acetylation; SV40 LT antigen also interacts with CBP/p300. It is perhaps not advisable to study TPX2 function and the acetylation of TPX2 in these cells.
6. Fig S5A: The IPs and inputs should be displayed on the same gel.
7. Fig 5D: The ability of CBP to acetylate TPX2, relative to the TPX2 RRR mutant, should be tested.
8. Fig 6: Graph mis-labelled. Tubulin not actin.

Reviewer #2 (Remarks to the Author):

The manuscript from Chao-Chieh Lin et al reports about the effect of COASY knockdown on

coenzyme A and Acetyl-Coenzyme A synthesis as well as protein acetylation. Special attention is devoted to the regulation of the protein acetyl transferase CBP and its role during mitosis. The authors claim that an increase of protein acetylation has been observed in a reduced COASY environment followed by CBP hyperacetylation and activation.

Major points:

This work is well written well technically performed and of potential interest. However, it requires much more work before it can be further considered.

1) The author spend a lot of effort demonstrating in different cellular contexts that COASY KD reduces CoA and ACoA synthesis while increasing acetylation on a subset fo acetyltable proteins. This is an interesting onservation although the authors do not provide evidence about how the reduced ACoA synthesis may led to increased protein acetylation. A number of congtrrol experiments should be considered including measurement of mitochondrial AcoA concentration. This may be of interest in consequence of the evidence that AcoA highest concentration resides in mitochondria and that most of the mitochondrial proteins are acetylated.

2) CP a nd P300 share aout 63% sequence homology. However the authors do not provide evidences about the effect of COASY KD on p300. It would of interest to understand whether COASY KD affects also p300 or it determines a CBP specific functional impairment. Does COASY binds p300?

3) This is a question that may have more general consequences. All lysine acetylases rely on ACoA availability. Does COASY KD affects acetylases other that CBP? For examples that of the P300-CBP associated factor (PCAF)?

4) The association COASY-CBP-TPX2 is analysed by co-immunoprecipitation experiments. However, due to the novelty of the COASY-CBP physical interation mass-spect methodologies could also be implemented to confirm this evidence.

5) The most crucial question, however, remains whether the COASY-CBP interation occurs in an in vivo context. The authors should explore this interaction at least in resting or TPA or PMA activated perypheral blood monocytes. This experiment may help to understand whether the mechanism described in this work may have any relevance during pathophysiological processes such as inflammation/infection.

Minor points:

a) fluorescense panels A and B in figure 5 are not convincing. Crucial controls are missing including the expression pattern of a GFP alone expression vector.

Reviewer #3 (Remarks to the Author):

Lin et al investigate importance of Coenzyme A Synthase (COASY)-controlled acetylation for mitotic progression. They find that cells depleted of COASY need longer to complete mitosis and display a high rate of cytokinesis failure and multinucleation (~60%). Despite the reduction of acetyl-CoA levels in COASY-depleted cells, comparison of the acetylated proteome in control and COASY-depleted cells revealed a group of hyperacetylated proteins involved in mitotic progression, including KIF23 (Mklp1) and the Aurora A activator TPX2 in COASY depleted cells. Lin et al . then pursue TPX2 as a potential target for COASY regulation. They show that TPX2 hyperacetylation is carried out by CREB-binding protein (CBP) which is negatively regulated by COASY.

Hyperacetylated TPX2 is protected from ubiquitin-mediated degradation and accumulates in COASY depleted cells potentially causing the mitotic defects. In line with a critical for CBP in regulating TPX2, depletion of CBP rescues the multinucleation phenotype (Figure 6B).

The manuscript highlights the importance of acetylation as an important post-translational modification for mitotic progression and should be of interest to a wide readership, however, there are some problems with this study. The first one is that it is unclear why COASY depletion results in protein hyperacetylation despite the clear expectation of the opposite. The explanation offered is that COASY adopts a different, non-catalytic role during mitosis which involves inhibiting CBP. The strongest evidence for this idea is that a COASY construct with a mutation abrogating catalytic activity rescues the mitotic phenotype (Figure 6B). In the light of technical concerns regarding the high variability of the reported levels of multinucleation in RNAi and RNAi rescue experiments in this study (see Minor points), this piece of evidence is not completely convincing, though, and requires strengthening.

The second somewhat unexpected turn in this manuscript is that for further analysis the authors focus on TPX2, not KIF23, which was also identified in their mass spec analysis, when a phenotype of 60% micronucleation would clearly point to an involvement of bona fide cytokinesis regulators such as KIF23/Mklp1. There may be perfectly good reasons for why the authors focused on TPX2 but this is not made clear in the manuscript. Did the authors test KIF23 and found it not to be a target, or are there other reasons for not considering KIF23 as a key candidate target of COASY depletion?

The analysis of TPX2 as such is nicely conducted and shows that acetylation of TPX2 by CBP interferes with normal ubiquitylation and degradation of TPX2. However, the analysis of TPX2 is then not brought back to the cellular phenotype. Why does hyperacetylation and hyperstabilization of TPX2 interfere with mitotic progression and cell division? Can the phenotype be rescued by a TPX2 mutant that cannot be acetylated? What do cells look like in immunofluorescence analysis when COASY is depleted? Is there more TPX2 and phospho-Aurora A on the spindle in comparison to control cells? There are clearly a lot of questions left, some of which could fairly easily be answered and would significantly advance the manuscript.

Altogether, currently this manuscript does not entirely come together. The part of the story relating to TPX2 regulation by CBP is itself a well conducted piece of work but does not tie in with the rest.

Minor points:

- There are no Western blots for the RNAi rescue experiments. It has to be demonstrated that the endogenous protein was depleted efficiently and the transgenes were expressed at comparable levels to endogenous protein and to each other.
- Some of the data are not consistent. Why do the authors observe 60% cytokinesis failure but only 20% multinucleation in COASY depleted cells (Figure 1C and F)? There also seems to be a high variation in the levels of multinucleation. In Figure 3E 35% multinucleation is depicted as a phenotype whereas in figure 6B a similar level of multinucleation is regarded as rescue.
- Figure 3F: If Figure 3F depicts a wash-out from Nocodazole into fresh medium, why has time point 0 much less Aurora A and phospho-Aurora A than the following time points when 0 should be the mitotic peak?
- Figure 5C: The immunoprecipitation experiment does not have an appropriate control. The Ig control alone is not sufficient. It has to be demonstrated that under the conditions used a specific immunoprecipitation was achieved, and that not the entire cell was precipitated. The immunoprecipitations, not just the inputs, should have been blotted for cyclin B and tubulin as well to demonstrate that these proteins do not co-precipitate. The same is true for Figures S4C and S5. Without demonstrating that there are proteins that do NOT co-precipitate under the conditions used, the data don't have much value.

We want to thank our reviewers for their careful reading of our manuscript and helpful suggestions for improvement. Please find the detailed point-to-point response below.

Reviewers' comments:

Reviewer #1 (Remarks to the Author):

The manuscript by Lin and colleagues presents data to suggest that Coenzyme A synthase (COASY) inactivation, perhaps paradoxically, leads to the hyper-acetylation of a network of proteins involved in mitotic regulation; COASY knockdown cells display prolonged mitoses and multinucleation. The authors do not offer any mechanistic insight into the relationship between COASY knockdown and hyper-acetylation, but focus their efforts on discerning the role of acetylation in mitotic progression.

Response:

Intuitively, knockdown of COASY should reduce the production of Coenzyme A, lower acetyl-CoA level, and cause global protein hypoacetylation. However, this simple concept is not supported by the experimental evidence. Several recent studies suggest that most histone acetyltransferases have similar affinity for CoA and acetyl-CoA. Therefore, CoA can competitively inhibit acetyl-CoA production. Therefore, the acetyl-CoA/CoA ratio, instead of absolute acetyl-CoA levels, may be more relevant determinants of protein acetylation^{1,2}. In our analysis of COASY-knocked down cells, while both CoA and acetyl-CoA are lower (Supplementary Fig. 1c, d), the acetyl-CoA/CoA ratio remain unchanged (Supplementary Fig. 1e). Consistently, our acetylome analysis revealed that the majority (89%, 955 acetylated /1074 total) of the acetylated peptides were not significantly altered by COASY knockdown (Fig. 2b). These results are consistent with other report that decreasing CoA level by inhibition of PANK, the upstream enzyme of COASY in CoA biosynthesis pathway, also did not alter the global pattern of protein acetylation³. Instead, COASY knockdown altered the acetylation of a specific set of proteins, including CBP and TPX2. These unexpected findings led us to identify a novel regulatory role of COASY in mitosis through stage-specific binding and regulating CBP acetyltransferase activity instead of acetyl-CoA or CoA levels. To gain further insight on how COASY might affect CBP activity, in the previous manuscript, we have provided multiple lines of supporting evidence that COAY regulates the TPX2 acetylation via a stage-specific interaction with CBP during mitosis (Fig. 5a-c). Once associated, COASY inhibits CBP and TPX2 acetylation to allow the TPX2 degradation, Aurora A inactivation and mitotic exits (Fig. 3f). These experiments include the tight temporal association between CBP-COASY interactions with TPX2 inactivation (Fig. 5c), the CBP-interacting domain (PPAT) of COASY is necessary and sufficient for reversing the multi-nucleation phenotypes (Fig. 6c) and the DPCK enzymatic dead mutant did not affect the COASY to rescue multi-nucleation phenotypes of COASY knockdown (Fig. 6a, b). COASY was able to inhibit the CBP HAT activities during in vitro biochemical assays (Fig 5f).

In response to the comments of the reviewers, we have performed additional experiments to support the concept that COASY regulated CBP independent of CoA synthesis pathways. We

knocked down multiple enzymes in the CoA biosynthesis pathway, including PANK and PPCDC (Supplemental Fig. 6d-g). None of these treatments were unable to induce TPX2 upregulation and multinucleation (Supplemental Fig. 6h, i) seen with COASY knockdown. These experiments further strengthened that the CoA and acetyl-CoA levels are not responsible for the multi-nucleation phenotypes observed with COASY knockdown. Therefore, we hypothesized that the novel and non-canonical regulatory role of COASY in CBP mediated acetylation may partially account for the hyperacetylation of the proteins found in the acetylome analysis. We have further clarified these concepts in the revised manuscript.

The authors present evidence to indicate that CBP acetylates and stabilizes the TPX2 protein, which in turn leads to the hyper-activation of Aurora A kinase, whose over-expression has previously been shown to promote cytokinesis failure and multinucleation (Cancer Cell, 2003 Jan;3(1):51-62.). In this regard the authors determine that COASY, CBP and TPX2 associate temporally at the mitotic spindle during mitosis and that the PPAT domain of COASY, which interacts with CBP, will rescue the multinucleation phenotype. The data presented is novel, and will be of interest to the wider scientific community.

Response:

We appreciate the interests from the reviewer and concur that our novel findings will be of interest to the wider scientific community.

Specific comments:

1. Although the authors present some evidence to suggest that TPX2 knockdown rescues the COASY knockdown-induced multinucleation phenotype and prolonged mitosis phenotype (Fig 3E and F), The manuscript could be strengthened by testing the ability of an siRNA-resistant wild-type TPX2 and a TPX2 RRR mutant, which is resistant to CBP-directed acetylation, to rescue the prolonged mitoses, and/or multinucleation phenotype when the endogenous TPX2 species is knocked down by siRNA. It would also be interesting to see if a TPX2 QQQ acetylation mimic mutant, reinforces these phenotypes.

Response:

In the revised manuscript, we have used a siRNA targeting the 3' UTR of TPX2 to lower the endogenous TPX2 under COASY knockdown, then replaced it with a wild-type, acetylation-deficient or acetylation-mimicking TPX2 mutants cDNA without the 3' UTR. As shown in revised Fig 3e and Supplementary Fig. S3m-n, wild-type and acetylation-mimicking mutant TPX2 expression increased multinucleation and extended mitosis, while acetylation-deficient mutant significantly reduced the ability of TPX2 to cause this phenotype.

2. The authors present evidence of a TPX2 acetylation/ubiquitylation switch, such that the TPX2-RRR mutant which is unable to be acetylated by CBP, is also targeted for ubiquitylation (Figure S4). It appears that the ubiquitylation studies were performed by IP-WB, under native conditions. Although the results are suggestive of ubiquitylation of TPX2 at the K residues identified, these experiments should be

repeated with His6-Ubiquitin under denaturing conditions to validate that it is TPX2 itself that is the ubiquitylation target.

Response:

Following reviewer's suggestion, we have repeated the analysis with HA-Ubiquitin under denaturing condition (1% SDS and boiling for 5min). The results are similar to previous results under native condition. These data have been presented in Supplementary Fig. 4h of the revised manuscript.

3. Fig S1: Did siCOASY #3 affect acetyl-CoA levels or multinucleation?

Response:

We have performed the analysis and confirmed that siCOASY# 3 also induced multinucleation (Supplementary Fig. 1f).

4. Fig 3C. The increase in TPX2 acetylation by WB using a pan-AcK antibody following COASY knockdown is not convincing. It would be valuable to generate a site-specific AcK antibody for TPX2, against one, or more of the sites.

Response:

We agree with the reviewers that it will be helpful to further validate the increased acetylation of TPX2. However, our previous experience with multiple projects have revealed that acetylated peptides, unlike phosphorylated peptides, do not always induce specific immune response for site-specific AcK antibody. Therefore, we have used an alternative specific quantitative approach using mass spectrometry to perform Parallel-Reaction Monitoring (PRM) with the doped heavy synthetic acetylated-peptides for quantification of the TPX2 acetylated peptides. These data have been presented in Supplementary Fig. 3g-l of the revised manuscript.

5. A number of experiments in this manuscript are performed with HEK-293T cells. It is well established that adenovirus E1A associates with CBP and p300 to reprogramme cellular acetylation; SV40 LT antigen also interacts with CBP/p300. It is perhaps not advisable to study TPX2 function and the acetylation of TPX2 in these cells.

Response:

We wished to first clarify that the multinucleation phenotypes of COASY knockdown were reproduced in multiple cell lines (A549, MDA-MB231, PANC-1 and ARPE-19) that do not express T-antigen. We have also repeated several key experiments in either A549 or MDA-MB-231 cell line (Supplementary Fig. 4d, e, h). The repeated results are all consistent with the results performed in HEK-293T cells. Although viral oncoprotein large T does bind and inhibit transcriptional co-activator activity of CBP/CBP, it remains uncertain whether it actually inhibits CBP/p300 acetyltransferase activity. Experimentally, HEK-293T cells have been used extensively to study CBP and p300-mediated acetylation by many laboratories. We have

previously used this cell line to investigate p53, Hsp90 and cortactin acetylation by CBP and those studies were accepted for publication.

6. *Fig S5A: The IPs and inputs should be displayed on the same gel.*

Response:

In the revised manuscript, we have displayed the IPs and inputs on the same gels in Supplementary Fig. 5b.

7. *Fig 5D: The ability of CBP to acetylate TPX2, relative to the TPX2 RRR mutant, should be tested.*

Response:

In the revised manuscript, we have provided the results showing TPX2 RRR mutant abolished most of acetylated-lysine signal when incubated with CBP (Supplementary Fig. 5g).

8. *Fig 6: Graph mis-labelled. Tubulin not actin.*

Response:

We have corrected the labels.

Reviewer #2 (Remarks to the Author):

The manuscript from Chao-Chieh Lin et al reports about the effect of COASY knockdown on coenzyme A and Acetyl-Coenzyme A synthesis as well as protein acetylation. Special attention is devoted to the regulation of the protein acetyl transferase CBP and its role during mitosis. The authors claim that an increase of protein acetylation has been observed in a reduced COASY environment followed by CBP hyperacetylation and activation.

Major points:

This work is well written well technically performed and of potential interest. However, it requires much more work before it can be further considered.

1) The author spend a lot of effort demonstrating in different cellular contexts that COASY KD reduces CoA and ACoA synthesis while increasing acetylation on a subset of acetyltable proteins. This is an interesting observation although the authors do not provide evidence about how the reduced ACoA synthesis may led to increased protein acetylation. A number of control experiments should be considered including measurement of mitochondrial AcoA concentration. This may be of interest in consequence of the evidence that AcoA highest concentration resides in mitochondria and that most of the mitochondrial proteins are acetylated.

Response:

We have shown that COASY is temporally associated with CBP during specific stages in mitosis to repress its HAT activities, resulting in TPX2 inactivation and mitotic exits. Since this regulation of CBP is not related to the levels of CoA and acetyl-CoA and most regulation occurs at the cytosol during mitosis, we feel that measurement of mitochondria CoA levels may not be relevant for the investigated biology of TPX2 and mitosis progression. As reviewer has correctly pointed out, the level of acetyl-CoA level in mitochondria positively regulated mitochondrial acetylation⁴. However, in our acetylome analysis, we identified 81 acetylated peptides from 46 mitochondrial proteins. The acetylation of majority of these mitochondria proteins are not altered by COASY knockdown. We also provided confocal microscopy showing subcellular localization of COASY, CBP and TPX2 during mitosis (Fig. 5a, b).

Intuitively, knockdown of COASY should reduce the production of Coenzyme A, lower acetyl-CoA level, and cause global protein hypoacetylation. However, this simple concept is not supported by the experimental evidence. Several recent studies suggest that most histone acetyltransferases have similar affinity for CoA and acetyl-CoA. Therefore, CoA can competitively inhibit acetyl-CoA production. Therefore, the acetyl-CoA/CoA ratio, instead of absolute acetyl-CoA levels, may be more relevant determinants of protein acetylation^{1, 2}. In our analysis of COASY-knocked down cells, while both CoA and acetyl-CoA are lower (Supplementary Fig. 1c, d), the acetyl-CoA/CoA ratio remain unchanged (Supplementary Fig. 1e). Consistently, our acetylome analysis revealed that the majority (89%, 955 acetylated /1074 total) of the acetylated peptides were not significantly altered by COASY knockdown (Fig. 2b). These results are consistent with other report that decreasing CoA level by inhibition of PANK, the upstream enzyme of COASY in CoA biosynthesis pathway, also did not alter the global pattern of protein acetylation³. Instead, COASY knockdown altered the acetylation of a specific set of proteins, including CBP and TPX2. These unexpected findings led us to identify a novel regulatory role of COASY in mitosis through stage-specific binding and regulating CBP acetyltransferase activity instead of acetyl-CoA or CoA levels. To gain further insight on how COASY might affect CBP activity, in the previous manuscript, we have provided multiple lines of supporting evidence that COASY regulates the TPX2 acetylation via a stage-specific interaction with CBP during mitosis (Fig. 5a-c). Once associated, COASY inhibits CBP and TPX2 acetylation to allow the TPX2 degradation, Aurora A inactivation and mitotic exits (Fig. 3f). These experiments include the tight temporal association between CBP-COASY interactions with TPX2 inactivation (Fig. 5c), the CBP-interacting domain (PPAT) of COASY is necessary and sufficient for reversing the multi-nucleation phenotypes (Fig. 6c) and the DPCCK enzymatic dead mutant did not affect the COASY to rescue multi-nucleation phenotypes of COASY knockdown (Fig. 6a, b). COASY was able to inhibit the CBP HAT activities during in vitro biochemical assays (Fig 5f).

In response to the comments of the reviewers, we have performed additional experiments to support the concept that COASY regulated CBP independent of CoA synthesis pathways. We knocked down multiple enzymes in the CoA biosynthesis pathway, including PANK and PPCDC (Supplemental Fig. 6d-g). None of these treatments were unable to induce TPX2 upregulation and multinucleation (Supplemental Fig. 6h, i) seen with COASY knockdown. These experiments further strengthened that the CoA and acetyl-CoA levels are not responsible

for the multi-nucleation phenotypes observed with COASY knockdown. Therefore, we hypothesized that the novel and non-canonical regulatory role of COASY in CBP mediated acetylation may partially account for the hyperacetylation of the proteins found in the acetylome analysis. We have further clarified these concepts in the revised manuscript.

2) CP and P300 share about 63% sequence homology. However the authors do not provide evidences about the effect of COASY KD on p300. It would of interest to understand whether COASY KD affects also p300 or it determines a CBP specific functional impairment. Does COASY binds p300?

Response:

Since CBP and P300 share sequence homology and substrates, we knocked down both COASY and p300 by targeting p300-specific sequences to examine whether repressing p300 can rescue COASY dependent multinucleation. The results showed that p300 knockdown did not rescue COASY-depleted multinucleation (Supplementary Fig. 4l-n), while CBP knockdown did. We also found 24 acetylated peptides of P300 in our acetylome analysis. However, none of them were affected by COASY knockdown. These data indicate that this regulatory mechanism of COASY to be CBP specific.

3) This is a question that may have more general consequences. All lysine acetylases rely on ACoA availability. Does COASY KD affect acetylases other that CBP? For examples that of the P300-CBP associated factor (PCAF)?

Response:

There are many ways various HATs are regulated independent of substrate (acetyl-CoA) concentrations. As our mechanisms are likely to occur through COASY-CBP association instead of general changes in the CoA and Acetyl-CoA levels, we may not expected to see a global change in all HATs. In our acetylome profiling data, COASY knockdown increase the acetylation of many members of the HBO1 complexes. Therefore, HBO1 may be another affected HAT upon COASY knockdown. We are actively working on this angle, but it will be beyond the scope of the current manuscript. PCAF may be another possible candidate HAT, but we currently don't have evidence for its involvement.

4) The association COASY-CBP-TPX2 is analyzed by co-immunoprecipitation experiments. However, due to the novelty of the COASY-CBP physical interaction mass-spec methodologies could also be implemented to confirm this evidence.

Response:

Following reviewer's suggestion, we have performed mass spectrometry and confirmed enrichment of COASY peptides when Flag-CBP was pulled down from HEK-293T cells (Supplementary Fig. 5c-f).

5) *The most crucial question, however, remains whether the COASY-CBP interaction occurs in an in vivo context. The authors should explore this interaction at least in resting or TPA or PMA activated peripheral blood monocytes. This experiment may help to understand whether the mechanism described in this work may have any relevance during pathophysiological processes such as inflammation/infection.*

Response:

We agree that it will be helpful to provide data on the in vivo relevance during pathophysiological processes. It is important to point out that we have shown the physiological relevance of COASY-CBP-TPX2 in the TPX2 degradation and mitotic exits. However, given that Aurora A and multi-nucleation is particularly relevant in cancer biology, we have provided two pieces of data with pathophysiological relevance in the revised manuscript. First, COASY knockdown leads to overexpression of TPX2 and Aurora A activation. Since Aurora A overexpression was reported to confer resistance to taxol⁵, we tested whether COASY knockdown can cause resistance to taxol (Supplementary Fig. 7a)⁵. We have found COASY knockdown in MDA-MB-231 cells did increase the taxol resistance, consistent with Aurora A activation. We further re-analyzed COASY expression level in human tumors using published expression datasets. We found that lower COASY expression in tumors is associated with metastasis and higher death rate (Supplementary Fig. 7b-d). These additional data provide the pathophysiological relevance of COASY in cancer-related setting.

Minor points:

a) fluorescence panels A and B in figure 5 are not convincing. Crucial controls are missing including the expression pattern of a GFP alone expression vector.

Response:

In the revised manuscript, we included GFP alone expression vector as negative control (Supplementary Fig. 5a), which showed diffuse pattern distinct from the COASY-GFP.

Reviewer #3 (Remarks to the Author):

Lin et al investigate importance of Coenzyme A Synthase (COASY)-controlled acetylation for mitotic progression. They find that cells depleted of COASY need longer to complete mitosis and display a high rate of cytokinesis failure and multinucleation (~60%). Despite the reduction of acetyl-CoA levels in COASY-depleted cells, comparison of the acetylated proteome in control and COASY-depleted cells revealed a group of hyperacetylated proteins involved in mitotic progression, including KIF23 (Mklp1) and the Aurora A activator TPX2 in COASY depleted cells. Lin et al . then pursue TPX2 as a potential target for COASY regulation. They show that TPX2 hyperacetylation is carried out by CREB-binding protein (CBP) which is negatively regulated by COASY. Hyperacetylated TPX2 is protected from ubiquitin-mediated degradation and accumulates in COASY depleted cells potentially causing the mitotic defects. In line with a critical for CBP in regulating TPX2, depletion of CBP rescues the multinucleation phenotype (Figure 6B).

The manuscript highlights the importance of acetylation as an important post-translational modification for mitotic progression and should be of interest to a wide readership, however, there are some problems with this study.

The first one is that it is unclear why COASY depletion results in protein hyperacetylation despite the clear expectation of the opposite. The explanation offered is that COASY adopts a different, non-catalytic role during mitosis which involves inhibiting CBP. The strongest evidence for this idea is that a COASY construct with a mutation abrogating catalytic activity rescues the mitotic phenotype (Figure 6B). In the light of technical concerns regarding the high variability of the reported levels of multinucleation in RNAi and RNAi rescue experiments in this study (see Minor points), this piece of evidence is not completely convincing, though, and requires strengthening.

Response:

We have addressed similar concerns in the response to reviewer #1, which is replicated here again.

Intuitively, knockdown of COASY should reduce the production of Coenzyme A, lower acetyl-CoA level, and cause global protein hypoacetylation. However, this simple concept is not supported by the experimental evidence. Several recent studies suggest that most histone acetyltransferases have similar affinity for CoA and acetyl-CoA. Therefore, CoA can competitively inhibit acetyl-CoA production. Therefore, the acetyl-CoA/CoA ratio, instead of absolute acetyl-CoA levels, may be more relevant determinants of protein acetylation^{1,2}. In our analysis of COASY-knocked down cells, while both CoA and acetyl-CoA are lower (Supplementary Fig. 1c, d), the acetyl-CoA/CoA ratio remain unchanged (Supplementary Fig. 1e). Consistently, our acetylome analysis revealed that the majority (89%, 955 acetylated /1074 total) of the acetylated peptides were not significantly altered by COASY knockdown (Fig. 2b). These results are consistent with other report that decreasing CoA level by inhibition of PANK, the upstream enzyme of COASY in CoA biosynthesis pathway, also did not alter the global pattern of protein acetylation³. Instead, COASY knockdown altered the acetylation of a specific set of proteins, including CBP and TPX2. These unexpected findings led us to identify a novel regulatory role of COASY in mitosis through stage-specific binding and regulating CBP acetyltransferase activity instead of acetyl-CoA or CoA levels. To gain further insight on how COASY might affect CBP activity, in the previous manuscript, we have provided multiple lines of supporting evidence that COASY regulates the TPX2 acetylation via a stage-specific interaction with CBP during mitosis (Fig. 5a-c). Once associated, COASY inhibits CBP and TPX2 acetylation to allow the TPX2 degradation, Aurora A inactivation and mitotic exits (Fig. 3f). These experiments include the tight temporal association between CBP-COASY interactions with TPX2 inactivation (Fig. 5c), the CBP-interacting domain (PPAT) of COASY is necessary and sufficient for reversing the multi-nucleation phenotypes (Fig. 6c) and the DPCK enzymatic dead mutant did not affect the COASY to rescue multi-nucleation phenotypes of COASY knockdown (Fig. 6a, b). COASY was able to inhibit the CBP HAT activities during in vitro biochemical assays (Fig 5f).

In response to the comments of the reviewers, we have performed additional experiments to support the concept that COASY regulated CBP independent of CoA synthesis pathways. We

knocked down multiple enzymes in the CoA biosynthesis pathway, including PANK and PPCDC (Supplemental Fig. 6d-g). None of these treatments were unable to induce TPX2 upregulation and multinucleation (Supplemental Fig. 6h, i) seen with COASY knockdown. These experiments further strengthened that the CoA and acetyl-CoA levels are not responsible for the multi-nucleation phenotypes observed with COASY knockdown. Therefore, we hypothesized that the novel and non-canonical regulatory role of COASY in CBP mediated acetylation may partially account for the hyperacetylation of the proteins found in the acetylome analysis. We have further clarified these concepts in the revised manuscript.

The second somewhat unexpected turn in this manuscript is that for further analysis the authors focus on TPX2, not KIF23, which was also identified in their mass spec analysis, when a phenotype of 60% micronucleation would clearly point to an involvement of bona fide cytokinesis regulators such as KIF23/Mklp1. There may be perfectly good reasons for why the authors focused on TPX2 but this is not made clear in the manuscript. Did the authors test KIF23 and found it not to be a target, or are there other reasons for not considering KIF23 as a key candidate target of COASY depletion?

Response:

We totally agree with reviewers that there are additional interesting candidates, including KIF23/MKLP1, to be tested. Indeed, KIF23/MKLP1 is a well-known protein for regulating cytokinesis. We did consider KIF23/MKLP1 as potential target. Here we provided data showing that COASY knockdown increased KIF23/MKLP1 protein expression. Knockdown of KIF23/MKLP1 alone dramatically increased the multi-nucleation. Importantly, KIF23/MKLP1 and COASY double knockdown did not rescue COASY dependent multinucleation. Since we looked for acetylated target protein responsible for COASY dependent mitotic defects, we did not follow up the functional role of KIF23/MKLP1 in the contexts of COASY knockdown.

The analysis of TPX2 as such is nicely conducted and shows that acetylation of TPX2 by CBP interferes with normal ubiquitination and degradation of TPX2. However, the analysis of TPX2 is then not brought back to the cellular phenotype.

Why does hyperacetylation and hyperstabilization of TPX2 interfere with mitotic progression and cell division?

Can the phenotype be rescued by a TPX2 mutant that cannot be acetylated?

Response:

In the revised manuscript, we have analyzed how various TPX2 acetylation mutants affect the multinucleation and extended mitosis. We used siRNA targeting 3' UTR of TPX2 to lower the endogenous TPX2, then replaced with wild-type, acetylation-deficient or acetylation-mimicking mutants of TPX2 cDNA, without the 3' UTR, to determine their ability to affect the phenotypes under COASY knockdown. As shown in revised Fig 3e and Supplemental Fig. S3m-n, wild-type and acetylation-mimicking mutant increased multinucleation and extended mitosis, while acetylation-deficient mutant rescued this phenotype.

What do cells look like in immunofluorescence analysis when COASY is depleted? Is there more TPX2 and phospho-Aurora A on the spindle in comparison to control cells?

Response:

In response to the reviewer's comments, we have performed additional immunofluorescence analysis. During metaphase in A549 cells, TPX2 was mostly localized in spindle poles (Supplementary Fig. 3o). Under COASY knockdown, we observed more TPX2 signal extending from spindle poles to chromosomes. During metaphase, the stage Aurora A are activated under both control and COASY knockdown (Fig. 3f, left panel of Western blots), the signal of Aurora A phosphorylation showed no obvious change under COASY knockdown (Supplementary Fig. 3p). However, during interphase, Aurora A phosphorylation was much higher under COASY knockdown, suggesting that the availability of COASY can determine the precise activation of Aurora A at specific stages of cell cycle (Supplementary Fig. 3q).

There are clearly a lot of questions left, some of which could fairly easily be answered and would significantly advance the manuscript.

Altogether, currently this manuscript does not entirely come together. The part of the story relating to TPX2 regulation by CBP is itself a well conducted piece of work but does not tie in with the rest.

Response:

In the revised manuscript, we have analyzed how various TPX2 mutants affect the multinucleation phenotypes. We are convinced that these additional work have significantly improved the manuscripts.

Minor points:

- *There are no Western blots for the RNAi rescue experiments. It has to be demonstrated that the endogenous protein was depleted efficiently and the transgenes were expressed at comparable levels to endogenous protein and to each other.*

Response:

In the revised manuscript, we have provided Western blots for the RNAi-rescued experiments in Supplementary Fig. 6b.

• *Some of the data are not consistent. Why do the authors observe 60% cytokinesis failure but only 20% multinucleation in COASY depleted cells (Figure 1C and F)? There also seems to be a high variation in the levels of multinucleation. In Figure 3E 35% multinucleation is depicted as a phenotype whereas in figure 6B a similar level of multinucleation is regarded as rescue.*

Response:

The variability of the specific percentage of cytokinesis and multi-nucleation comes from the specific timing and the balance between cytokinesis and cell death in the particular experiments. Importantly, all the experiments were done in parallel and the trends of altered multinucleation were highly consistent among different experiments. The event of cytokinesis was counted when the cells entered mitosis. For multinucleation, we fixed the samples for all the cells at specific moment. Also, cytokinesis failure may lead to cell death of certain cells and only a portion of the cells with cytokinesis failure will survive and manifest as multi-nucleated cells.

About variation in the level of multinucleation in figure 3e, we performed double siRNA transfection. To find a balance between cell toxicity of lipofectamine and optimal ratio of siRNA/lipofectamine, we chose to scale down the amount of siRNA for both siRNA to be transfected at the same time. Therefore, the effect of knockdown is more modest.

• *Figure 3F: If Figure 3F depicts a wash-out from Nocodazole into fresh medium, why has time point 0 much less Aurora A and phospho-Aurora A than the following time points when 0 should be the mitotic peak?*

Response:

TPX2 is involved in microtubule assembly and targeting Aurora A to microtubule and Aurora A stability^{6,7,8}. Since nocodazole interferes with polymerization of microtubule to arrest cells in prometaphase⁹, we think proper microtubule assembly may affect the Aurora A stability by interacting with TPX2. However, proving this hypothesis will be beyond the scope of the current manuscript.

• *Figure 5C: The immunoprecipitation experiment does not have an appropriate control. The Ig control alone is not sufficient. It has to be demonstrated that under the conditions used a specific immunoprecipitation was achieved, and that not the entire cell was precipitated. The immunoprecipitations, not just the inputs, should have been blotted for cyclin B and tubulin as well to demonstrate that these proteins do not co-precipitate. The same is true for Figures S4C and S5. Without*

demonstrating that there are proteins that do NOT co-precipitate under the conditions used, the data don't have much value.

Response: In the revised manuscript, we have repeated these experiments with additional suggested controls in Fig. 5c and Supplementary Fig. 4c and 5b.

References:

1. Pietrocola, F., Galluzzi, L., Bravo-San Pedro, J.M., Madeo, F. & Kroemer, G. Acetyl coenzyme A: a central metabolite and second messenger. *Cell Metab* **21**, 805-821 (2015).
2. Galdieri, L., Zhang, T., Rogerson, D., Lleshi, R. & Vancura, A. Protein acetylation and acetyl coenzyme a metabolism in budding yeast. *Eukaryot Cell* **13**, 1472-1483 (2014).
3. Siudeja, K. *et al.* Impaired Coenzyme A metabolism affects histone and tubulin acetylation in Drosophila and human cell models of pantothenate kinase associated neurodegeneration. *EMBO Mol Med* **3**, 755-766 (2011).
4. Weinert, B.T. *et al.* Acetylation dynamics and stoichiometry in *Saccharomyces cerevisiae*. *Mol Syst Biol* **10**, 716 (2014).
5. Anand, S., Penrhyn-Lowe, S. & Venkitaraman, A.R. AURORA-A amplification overrides the mitotic spindle assembly checkpoint, inducing resistance to Taxol. *Cancer Cell* **3**, 51-62 (2003).
6. Giubettini, M. *et al.* Control of Aurora-A stability through interaction with TPX2. *J Cell Sci* **124**, 113-122 (2011).
7. Gruss, O.J. *et al.* Chromosome-induced microtubule assembly mediated by TPX2 is required for spindle formation in HeLa cells. *Nat Cell Biol* **4**, 871-879 (2002).
8. Kufer, T.A. *et al.* Human TPX2 is required for targeting Aurora-A kinase to the spindle. *J Cell Biol* **158**, 617-623 (2002).
9. Jordan, M.A., Thrower, D. & Wilson, L. Effects of vinblastine, podophyllotoxin and nocodazole on mitotic spindles. Implications for the role of microtubule dynamics in mitosis. *J Cell Sci* **102 (Pt 3)**, 401-416 (1992).

Reviewer #1 (Remarks to the Author):

The authors have addressed most of my concerns in the revised manuscript. In response to the authors responses to my original review, and my evaluation of the new manuscript I have the additional comments.

1. The authors cite papers that suggest that CoA/acetyl-CoA ratios are a more accurate determinant of acetyltransferase activity, and hence, protein acetylation. Although this a relatively new concept for acetylation there are obvious precedents, such as GTP/GDP ratios as determinants of low-molecular weight G-protein activity. The authors suggest that the ability of COASY to inhibit the CBP-dependent acetylation of TPX2 and limit multi-nucleation is independent of its ability to synthesize CoA, and relies upon the ability of the PPAT domain of COASY to interact directly with CBP, and inhibit CBP acetyltransferase in a non-canonical, i.e. not determined, manner (Figure 6). The authors also present data to suggest that knockdown of proteins involved in CoA synthesis (PANK and PPCDC) do not phenocopy the effects of COASY knockdown on TPX2 stabilization and multi-nucleation (Figure S6).

The crucial questions here are whether, in this experimental situation, knockdown of PANK or PPCDC affects the cellular CoA/acetyl-CoA ratio, and how these cellular CoA/acetyl-CoA ratios compare with the cellular CoA/acetyl-CoA ratios measured following COASY knockdown, and moreover, how PANK or PPCDC knockdown affects acetylation of CBP substrates in mitosis.

As such I think it is premature to suggest that non-canonical COASY activities are required for the inhibition of CBP in mitosis. As COASY is involved in the final reactions of CoA synthesis, it is possible that the PPAT domain of COASY could bind to CoA and facilitate loading of CoA on CBP to increase CBP CoA/acetyl-CoA ratios, or alternatively, facilitate the loading of the 3'-dephospho-CoA to CBP, to similarly reduce acetyl-CoA levels on CBP. Likewise, the R499C catalytic inactive mutant could possess CoA binding that effects CBP-directed acetyltransferase activity through the modulation of CoA/acetyl-CoA loading on CBP. It could also be postulated that PANK and PPCDC do not possess CoA binding capacity and therefore cannot directly modulate CoA/acetyl-CoA loading on CBP.

2. The use of the TPX2-QQQ and TPX2-RRR mutants (Figure 3E and Figure S3) are supportive of a role of TPX2 acetylation in the multi-nucleation phenotype.

3. The IPs under denaturing conditions (Figure S4h) support the idea that TPX2 is the ubiquitylation target at one, or more of the K residues identified. It would improve the manuscript if a longer exposure of the gel was shown to visualize the high molecular weight poly-ubiquitin smear, as at

present, only a band corresponding to mono-ubiquitylated (?) TPX2 species is seen- or is this TPX2 species the major form? If so this should be discussed in the manuscript.

4. I agree with the authors that site-specific AcK antibodies are difficult to generate. However, it would have been worth a try to generate this reagent.

5. I appreciate that the authors observe the COASY- multinucleation phenotype in cells that do not possess viral antigens. My comments however, regard the use of HEK-293T stands. The expression of adenovirus E1A and SV40-LT affects CBP function in these cells. Just because other labs have published using these cells to study CBP function does not make it right, or appropriate, to use these cells to study normal biological functions of CBP.

Reviewer #2 (Remarks to the Author):

The authors fulfilled my requests.

Reviewer #3 (Remarks to the Author):

The manuscript has been significantly improved by the revisions carried out by the authors, and I would now recommend publication. It may be a good idea, though, to incorporate the explanation of why KIF23 is not pursued as an alternative candidate to explain the multi nucleation phenotype into the manuscript (currently only found in the rebuttal letter)

We want to thank our reviewers for their careful reading of our revised manuscript and helpful suggestions for improvement. Please find the detailed point-to-point response to the review below.

Reviewer #1 (Remarks to the Author):

The authors have addressed most of my concerns in the revised manuscript. In response to the authors responses to my original review, and my evaluation of the new manuscript I have the additional comments.

1. The authors cite papers that suggest that CoA/acetyl-CoA ratios are a more accurate determinant of acetyltransferase activity, and hence, protein acetylation. Although this a relatively new concept for acetylation there are obvious precedents, such as GTP/GDP ratios as determinants of low-molecular weight G-protein activity. The authors suggest that the ability of COASY to inhibit the CBP-dependent acetylation of TPX2 and limit multi-nucleation is independent of its ability to synthesize CoA, and relies upon the ability of the PPAT domain of COASY to interact directly with CBP, and inhibit CBP acetyltransferase in a non-canonical, i.e. not determined, manner (Figure 6). The authors also present data to suggest that knockdown of proteins involved in CoA synthesis (PANK and PPCDC) do not phenocopy the effects of COASY knockdown on TPX2 stabilization and multi-nucleation (Figure S6).

The crucial questions here are whether, in this experimental situation, knockdown of PANK or PPCDC affects the cellular CoA/acetyl-CoA ratio, and how these cellular CoA/acetyl-CoA ratios compare with the cellular CoA/acetyl-CoA ratios measured following COASY knockdown, and moreover, how PANK or PPCDC knockdown affects acetylation of CBP substrates in mitosis.

As such I think it is premature to suggest that non-canonical COASY activities are required for the inhibition of CBP in mitosis. As COASY is involved in the final reactions of CoA synthesis, it is possible that the PPAT domain of COASY could bind to CoA and facilitate loading of CoA on CBP to increase CBP CoA/acetyl-CoA ratios, or alternatively, facilitate the loading of the 3'-dephospho-CoA to CBP, to similarly reduce acetyl-CoA levels on CBP. Likewise, the R499C catalytic inactive mutant could possess CoA binding that effects CBP-directed acetyltransferase activity through the modulation of CoA/acetyl-CoA loading on CBP. It could also be postulated that PANK and PPCDC do not possess CoA binding capacity and therefore cannot directly modulate CoA/acetyl-CoA loading on CBP.

Response: We agree with reviewer's very insightful comment. Indeed, we cannot fully exclude the possible involvement of COASY catalytic activity in regulating CBP CoA loading even though the production of CoA *per se* is not responsible. Since COASY silencing did not change the ratio between CoA and Acetyl-CoA, so it may not be relevant to measure the changes in the CoA and acetyl-CoA when other genes in the CoA synthesis pathways are inhibited. Therefore, we have modified the text to give equal weighting to the consideration of both canonical and non-canonical functions of COASY in the regulation of CBP.

2. The use of the TPX2-*QQQ* and TPX2-*RRR* mutants (Figure 3E and Figure S3) are supportive of a role of TPX2 acetylation in the multi-nucleation phenotype.

Response: We appreciate that review has recognized our response to the previous review.

3. The IPs under denaturing conditions (Figure S4h) support the idea that TPX2 is the ubiquitylation target at one, or more of the K residues identified. It would improve the manuscript if a longer exposure of the gel was shown to visualize the high molecular weight poly-ubiquitin smear, as at present, only a band corresponding to mono-ubiquitylated (?) TPX2 species is seen- or is this TPX2 species the major form? If so this should be discussed in the manuscript.

Response: We appreciate the comments from the reviewer. The misunderstanding may result from our incomplete indication of the size markers. We have modified the figure S4H to include the migration of additional size markers in the original gel. The size ranges of the HA-Ub smear are quite large, extending from 100-160 KDa. This broad size distribution is further supported by the smear pattern of TPX2 Western blot. Therefore, we expect the apparent broad-range smear of HA-Ub still represent the poly-ubiquitinated species of TPX2. We have modified the figures and discussed this issue in the revised manuscript.

4. I agree with the authors that site-specific AcK antibodies are difficult to generate. However, it would have been worth a try to generate this reagent.

Response: We appreciate the utility of such reagents and have made plan to generate these site-specific AcK antibodies for our follow-up investigation. For the purpose of the validating increased TPX2 acetylation, we believe that the use of parallel-reaction monitoring (PRM)

targeted mass spectrometry using synthetic heavy isotope peptides should be much more quantitative and precise in the degree of acetylation for each modified residue.

5. I appreciate that the authors observe the COASY- multinucleation phenotype in cells that do not possess viral antigens. My comments however, regard the use of HEK-293T stands. The expression of adenovirus E1A and SV40-LT affects CBP function in these cells. Just because other labs have published using these cells to study CBP function does not make it right, or appropriate, to use these cells to study normal biological functions of CBP.

Response: We agreed with the reviewer that HEK-293T cells might not be ideal to assess CBP activity. In our revision, we have repeated several key experiments in A594 and MDA-MB231, which do not express E1A or large T, and observed similar phenotypes. Therefore, the conclusion of our study is independent of E1A or large T.

Reviewer #2 (Remarks to the Author):

The authors fulfilled my requests.

Response: We appreciate that review has recognized our response to the previous review.

Reviewer #3 (Remarks to the Author):

The manuscript has been significantly improved by the revisions carried out by the authors, and I would now recommend publication. It may be a good idea, though, to incorporate the explanation of why KIF23 is not pursued as an alternative candidate to explain the multi nucleation phenotype into the manuscript (currently only found in the rebuttal letter)

Response: We appreciate that review has recognized our response to the previous review. While we appreciate the kind suggestion to include the KIF23 data, but we are concerned that such data may distract readers from the main conclusions of the manuscript. I hope the reviewer would understand our intention to focus on the TPX2 in this manuscript.